# Steerable Partial Differential Operators for Equivariant Neural Networks

**Erik Jenner**[*]
University of Amsterdam
erik@ejenner.com

**Maurice Weiler**
University of Amsterdam
m.weiler.ml@gmail.com

## Abstract

Recent work in equivariant deep learning bears strong similarities to physics. Fields over a base space are fundamental entities in both subjects, as are equivariant maps between these fields. In deep learning, however, these maps are usually defined by convolutions with a kernel, whereas they are partial differential operators (PDOs) in physics. Developing the theory of equivariant PDOs in the context of deep learning could bring these subjects even closer together and lead to a stronger flow of ideas. In this work, we derive a $G$-steerability constraint that completely characterizes when a PDO between feature vector fields is equivariant, for arbitrary symmetry groups $G$. We then fully solve this constraint for several important groups. We use our solutions as equivariant drop-in replacements for convolutional layers and benchmark them in that role. Finally, we develop a framework for equivariant maps based on Schwartz distributions that unifies classical convolutions and differential operators and gives insight about the relation between the two.

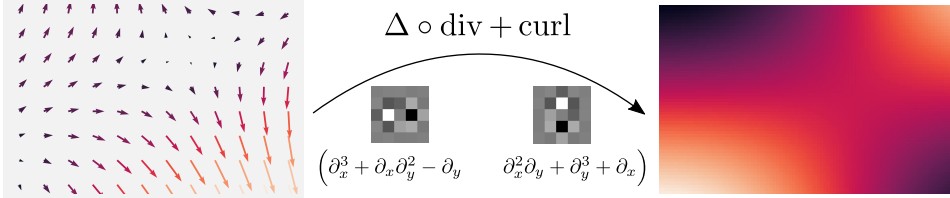

Figure 1: A vector field (left) can be mapped to a scalar field (right) by applying certain partial differential operators (PDOs), such as the Laplacian of the divergence and the 2D curl. Such a PDO from a 2D vector to a scalar field can be represented as a $2 \times 1$ matrix, where each of the two entries is a one-dimensional PDO that acts on one of the two components of the vector field. Similarly, matrices of PDOs with different dimensions map between other types of fields. Our goal is to find *all* PDOs for which this map becomes equivariant, for arbitrary types of fields. For the implementation, we will later discretize PDOs as stencils (middle).

## 1 Introduction

In many machine learning tasks, the data exhibits certain symmetries, such as translation- and sometimes rotation-invariance in image classification. To exploit those symmetries, equivariant neural networks have been widely studied and successfully applied in the past years, beginning with Group convolutional neural networks (Cohen & Welling, 2016; Weiler et al., 2018b). A significant generalization of Group convolutional networks is given by steerable CNNs (Cohen & Welling, 2017; Weiler et al., 2018a; Weiler & Cesa, 2019), which unify many different pre-existing equivariant models (Weiler & Cesa, 2019). They do this by representing features as fields of feature vectors over a base space, such as $\mathbb{R}^2$ in the case of two-dimensional images. Layers are then linear equivariant maps between these fields.

This is very reminiscent of physics. There, fields are used to model particles and their interactions, with physical space or spacetime as the base space. The maps between these fields are also equivariant, with the symmetries being part of fundamental physical laws.

---

[*]Work done during an internship at QUVA Lab

It is also noteworthy that these symmetries are largely ones that appear the most often in deep learning, such as translation and rotation equivariance. These similarities have already led to ideas from physics being applied in equivariant deep learning (Lang & Weiler, 2021).

However, one remaining difference is that physics uses equivariant partial differential operators (PDOs) to define maps between fields, such as the gradient or Laplacian. Therefore, using PDOs instead of convolutions in deep learning would complete the analogy to physics and could lead to even more transfer of ideas between subjects.

Equivariant PDO-based networks have already been designed in prior work (Shen et al., 2020; Smets et al., 2020; Sharp et al., 2020). Most relevant for our work are PDO-eConvs (Shen et al., 2020), which can be seen as the PDO-analogon of group convolutions. However, PDO-eConvs are only one instance of equivariant PDOs and do not cover the most common PDOs from physics, such as the gradient, divergence, etc. Very similarly to how steerable CNNs (Cohen & Welling, 2017; Weiler et al., 2018a; Weiler & Cesa, 2019) generalize group convolutions, we generalize PDO-eConvs by characterizing the set of *all* translation equivariant PDOs between feature fields over Euclidean spaces. Because of this analogy, we dub these equivariant differential operators *steerable PDOs*.

These steerable PDOs and their similarity to steerable CNNs also raise the question of how equivariant PDOs and kernels relate to each other, and whether they can be unified. We present a framework for equivariant maps that contains both steerable PDOs and convolutions with steerable kernels as special cases. We then prove that this framework defines the most general set of translation equivariant, linear, continuous maps between feature fields, complementing recent work (Aronsson, 2021) that describes when equivariant maps are convolutions. Since formally developing this framework requires the theory of Schwartz distributions, we cover it mainly in Appendix E, and the main paper can be read without any knowledge of distributions. However, we reference the main results from this framework in the paper where appropriate.

In order to make steerable PDOs practically applicable, we describe an approach to find complete bases for vector spaces of equivariant PDOs and then apply this method to the most important cases. We have also implemented steerable PDOs for all subgroups of $O(2)$ (`https://github.com/ejnnr/steerable_pdos`). Our code extends the E2CNN library[1] (Weiler & Cesa, 2019), which will allow practitioners to easily use both steerable kernels and steerable PDOs within the same library, and even to combine both inside the same network. Finally, we test our approach empirically by comparing steerable PDOs to steerable CNNs. In particular, we benchmark different discretization methods for the numerical implementation.

In summary, our main contributions are as follows:

- We develop the theory of equivariant PDOs on Euclidean spaces, giving a practical characterization of precisely when a PDO is equivariant under any given symmetry.
- We unify equivariant PDOs and kernels into one framework that provably contains all translation equivariant, linear, continuous maps between feature spaces.
- We describe a method for finding bases of the vector spaces of equivariant PDOs, and provide explicit bases for many important cases.
- We benchmark steerable PDOs using different discretization procedures and provide an implementation of steerable PDOs as an extension of the E2CNN library.

## 1.1 RELATED WORK

**Equivariant convolutional networks**   Our approach to equivariance follows the one taken by steerable CNNs (Cohen & Welling, 2017; Weiler et al., 2018a; Weiler & Cesa, 2019; Brandstetter et al., 2021). They represent each feature as a map from the base space, such as $\mathbb{R}^d$, to a fiber $\mathbb{R}^c$ that is equipped with a representation $\rho$ of the point group $G$. Compared to vanilla CNNs, which have fiber $\mathbb{R}$, steerable CNNs thus extend the *codomain* of feature maps.

A different approach is taken by group convolutional networks (Cohen & Welling, 2016; Hoogeboom et al., 2018; Weiler et al., 2018b). They represent each feature as a map from a group $H$ acting on the input space to $\mathbb{R}$. Because the input to the network usually does not lie in $H$, this requires a lifting

---

[1]`https://quva-lab.github.io/e2cnn/`

map from the input space to $H$. Compared to vanilla CNNs, group convolutional networks can thus be understood as extending the *domain* of feature maps.

When $H = \mathbb{R}^d \rtimes G$ is the semidirect product of the translation group and a pointwise group $G$, then group convolutions on $H$ are equivalent to $G$-steerable convolutions with regular representations. For finite $G$, the group convolution over $G$ simply becomes a finite sum. LieConvs (Finzi et al., 2020) describe a way of implementing group convolutions even for infinite groups by using a Monte Carlo approximation for the convolution integral. Steerable CNNs with regular representations would have to use similar approximations for infinite groups, but they can instead also use (non-regular) finite-dimensional representations. Both the group convolutional and the steerable approach can be applied to non-Euclidean input spaces—LieConvs define group convolutions on arbitrary Lie groups and steerable convolutions can be defined on Riemannian manifolds (Cohen et al., 2019b; Weiler et al., 2021) and homogeneous spaces (Cohen et al., 2019a).

One practical advantage of the group convolutional approach employed by LieConvs is that it doesn't require solving any equivariance constraints, which tends to make implementation of new groups easier. They also require somewhat less heavy theoretical machinery. On the other hand, steerable CNNs are much more general. This makes them interesting from a theoretical angle and also has more practical advantages; for example, they can naturally represent the symmetries of vector field input or output. Since our focus is developing the theory of equivariant PDOs and the connection to physics, where vector fields are ubiquitous, we are taking the steerable perspective in this paper.

**Equivariant PDO-based networks**   The work most closely related to ours are PDO-eConvs (Shen et al., 2020), which apply the group convolutional perspective to PDOs. Unlike LieConvs, they are not designed to work with infinite groups. The steerable PDOs we introduce generalizes PDO-eConvs, which are obtained as a special case by using regular representations.

A different approach to equivariant PDO-based networks was taken by Smets et al. (2020). Instead of applying a differential operator to input features, they use layers that map an initial condition for a PDE to its solution at a fixed later time. The PDE has a fixed form but several learnable parameters and constraints on these parameters—combined with the form of the PDE—guarantee equivariance. Sharp et al. (2020) also use a PDE, namely the diffusion equation, as part of their DiffusionNet model, which can learn on 3D surfaces. Interestingly, the time evolution operator for the diffusion equation is $\exp(t\Delta)$, which can be interpreted as an infinite power series in the Laplacian, very reminiscent of the finite Laplacian polynomials that naturally appear throughout this paper. Studying the equivariance of such infinite series of PDOs might be an interesting direction for future work. We clarify the relation between PDO-based and kernel-based networks in some more detail in Appendix F.

## 2   STEERABLE PDOs

In this section, we develop the theory of equivariant PDOs. We will represent all features as smooth fields $f : \mathbb{R}^d \to \mathbb{R}^c$ that associate a feature vector $f(x) \in \mathbb{R}^c$, called the *fiber* at $x$, with each point $x \in \mathbb{R}^d$. We write $\mathcal{F}_i = C^\infty(\mathbb{R}^d, \mathbb{R}^{c_i})$ for the space of these fields $f$ in layer $i$. Additionally, we have a group of transformations acting on the input space $\mathbb{R}^d$, which describes under which symmetries we want the PDOs to be equivariant. We will always use a group of the form $H = (\mathbb{R}^d, +) \rtimes G$, for some $G \leq \mathrm{GL}(d, \mathbb{R})$. Here, $(\mathbb{R}^d, +)$ refers to the group of translations of $\mathbb{R}^d$, while $G$ is some group of linear invertible transformations.

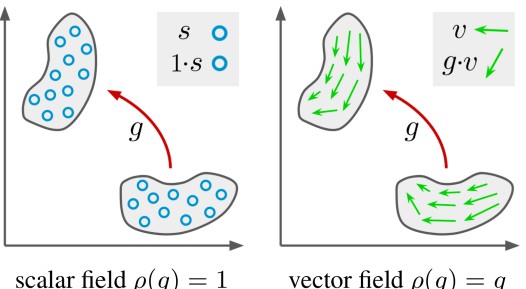

Figure 2: Transformation of scalar and vector fields (reproduced with permission from Weiler & Cesa (2019))

The full group of symmetries $H$ is the semidirect product of these two, meaning that each element $h \in H$ can be uniquely written as $h = tg$, where $t \in \mathbb{R}^d$ is a translation and $g \in G$ a linear transformation. For example, if $G = \{e\}$ is the trivial group, we consider only equivariance under translations, as in classical CNNs, while for $G = \mathrm{SO}(d)$ we additionally consider rotational equivariance.

Each feature space $\mathcal{F}_i$ has an associated group representation $\rho_i : G \to \mathrm{GL}(c_i, \mathbb{R})$, which determines how each fiber $\mathbb{R}^{c_i}$ transforms under transformations of the input space. Briefly, $\rho_i$ associates an invertible matrix $\rho_i(g)$ to each group element $g$, such that $\rho_i(g)\rho_i(g') = \rho_i(gg')$; more details on representation theory can be found in Appendix B. To see why these representations are necessary, consider the feature space $\mathcal{F} = C^\infty(\mathbb{R}^2, \mathbb{R}^2)$ and the group $G = \mathrm{SO}(2)$. The two channels could simply be two independent scalar fields, meaning that rotations of the input move each fiber but do not transform the fibers themselves. Formally, this would mean using *trivial representations* $\rho(g) = 1$ for both channels. On the other hand, the two channels could together form a vector field, which means that each fiber would need to itself be rotated in addition to being moved. This would correspond to the representation $\rho(g) = g$. These two cases are visualized in Fig. 2.

In general, the transformation of a feature $f \in \mathcal{F}_i$ under an input transformation $tg$ with $t \in \mathbb{R}^d$ and $g \in G$ is given by

$$((tg) \rhd_i f)(x) := \rho_i(g) f(g^{-1}(x - t)). \tag{1}$$

The $g^{-1}(x - t)$ term moves each fiber spatially, whereas the $\rho_i(g)$ is responsible for the individual transformation of each fiber.

For a network, we will need maps between adjacent feature spaces $\mathcal{F}_i$ and $\mathcal{F}_{i+1}$. Since during this theory section, we only consider single layers in isolation, we will drop the index $i$ and simply denote the layer map as $\Phi : \mathcal{F}_{\mathrm{in}} \to \mathcal{F}_{\mathrm{out}}$. We are particularly interested in *equivariant* maps $\Phi$, i.e. maps that commute with the action of $H$ on the feature spaces:

$$\Phi(h \rhd_{\mathrm{in}} f) = h \rhd_{\mathrm{out}} \Phi(f) \quad \forall h \in H, f \in \mathcal{F}_{\mathrm{in}}. \tag{2}$$

We call $\Phi$ *translation*-equivariant if Eq. (2) holds for $h \in (\mathbb{R}^d, +)$, i.e. for pure translations. Analogously, $\Phi$ is $G$-equivariant if it holds for linear transformations $h \in G$. Because $H$ is the semidirect product of $\mathbb{R}^d$ and $G$, a map is $H$-equivariant if and only if it is both translation- and $G$-equivariant.

## 2.1 PDOs as maps between feature spaces

We want to use PDOs for the layer map $\Phi$, so we need to introduce some notation for PDOs between multi-dimensional feature fields. As shown in Fig. 1, such a multi-dimensional PDO can be interpreted as a *matrix* of one-dimensional PDOs. Specifically, a PDO from $C^\infty(\mathbb{R}^d, \mathbb{R}^{c_{\mathrm{in}}})$ to $C^\infty(\mathbb{R}^d, \mathbb{R}^{c_{\mathrm{out}}})$ is described by a $c_{\mathrm{out}} \times c_{\mathrm{in}}$ matrix. For example, the 2D divergence operator, which maps from $\mathbb{R}^2$ to $\mathbb{R} = \mathbb{R}^1$ can be written as the $1 \times 2$ matrix $(\partial_1 \quad \partial_2)$. This is exactly analogous to convolutional kernels, which can also be interpreted as $c_{\mathrm{out}} \times c_{\mathrm{in}}$ matrices of scalar-valued kernels.

To work with the one-dimensional PDOs that make up the entries of this matrix, we use multi-index notation, so for a tuple $\alpha = (\alpha_1, \ldots, \alpha_d) \in \mathbb{N}_0^d$, we write $\partial^\alpha := \partial_1^{\alpha_1} \ldots \partial_d^{\alpha_d}$. A general one-dimensional PDO is a sum $\sum_\alpha c_\alpha \partial^\alpha$, where the coefficients $c_\alpha$ are smooth functions $c_\alpha : \mathbb{R}^d \to \mathbb{R}$ (so for now no spatial weight sharing is assumed). The sum ranges over all multi-indices $\alpha$, but we require all but a finite number of coefficients to be zero everywhere, so the sum is effectively finite. As described, a PDO between general feature spaces is then a matrix of these one-dimensional PDOs.

## 2.2 Equivariance constraint for PDOs

We now derive a complete characterization of the PDOs that are $H$-equivariant in the sense defined by Eq. (2). Because a map is equivariant under the full symmetries $H = (\mathbb{R}^d, +) \rtimes G$ if and only if it is both translation equivariant and $G$-equivariant, we split up our treatment into these two requirements.

First, we note that translation equivariance corresponds to spatial weight sharing, just like in CNNs (see Appendix G for the proof):

**Proposition 1.** *A $c_{out} \times c_{in}$-PDO $\Phi$ with matrix entries $\sum_\alpha c_\alpha^{ij} \partial^\alpha$ is translation equivariant if and only if all coefficients $c_\alpha^{ij}$ are constants, i.e. $c_\alpha^{ij}(x) = c_\alpha^{ij}(x')$ for all $x, x' \in \mathbb{R}^d$.*

So from now on, we restrict our attention to PDOs with constant coefficients and ask under which circumstances they are additionally equivariant under the action of the point group $G \le \mathrm{GL}(d, \mathbb{R})$. To answer that, we make use of a duality between polynomials and PDOs that will appear throughout this paper: for a PDO $\sum_\alpha c_\alpha \partial^\alpha$, with $c_\alpha \in \mathbb{R}$,[2] there is an associated polynomial simply given by

---

[2]Note that we had $c_\alpha \in C^\infty(\mathbb{R}^d, \mathbb{R})$ before, but we now restrict ourselves to constant coefficients and identify constant functions $\mathbb{R}^d \to \mathbb{R}$ with real numbers.

$\sum_\alpha c_\alpha x^\alpha$, where $x^\alpha := x_1^{\alpha_1} \ldots x_d^{\alpha_d}$. Conversely, for any polynomial $p = \sum_\alpha c_\alpha x^\alpha$, we get a PDO by formally plugging in $\partial = (\partial_1, \ldots, \partial_d)$ for $x$, yielding $p(\partial) := \sum_\alpha c_\alpha \partial^\alpha$. We will denote the map in this direction by $D$, so we write $D(p) := p(\partial)$.

We can extend this duality to PDOs between multi-dimensional fields: for a matrix $P$ of polynomials, we define $D(P)$ component-wise, so $D(P)$ will be a matrix of PDOs given by $D(P)_{ij} := D(P_{ij})$. To avoid confusion, we will always denote polynomials by lowercase letters, such as $p, q$, and matrices of polynomials by uppercase letters, like $P$ and $Q$.

As a simple example of the correspondence between polynomials and PDOs, the Laplacian operator $\Delta$ is given by $\Delta = D(|x|^2)$. The gradient $(\partial_1, \ldots, \partial_d)^T$ is induced by the $d \times 1$ matrix $(x_1, \ldots, x_d)^T$ and the 3D curl ($d = 3$) is induced by the $3 \times 3$ matrix

$$P = \begin{pmatrix} 0 & -x_3 & x_2 \\ x_3 & 0 & -x_1 \\ -x_2 & x_1 & 0 \end{pmatrix} \quad \Longrightarrow \quad D(P) = \begin{pmatrix} 0 & -\partial_3 & \partial_2 \\ \partial_3 & 0 & -\partial_1 \\ -\partial_2 & \partial_1 & 0 \end{pmatrix}. \quad (3)$$

Finally, note that $D$ is a ring isomorphism, i.e. a bijection with $D(p + q) = D(p) + D(q)$ and $D(pq) = D(p) \circ D(q)$, allowing us to switch viewpoints at will.

We are now ready to state our main result on the $G$-equivariance of PDOs:

**Theorem 2.** *For any matrix of polynomials $P$, the differential operator $D(P)$ is $G$-equivariant if and only if it satisfies the* PDO $G$-steerability constraint,

$$P\left((g^{-1})^T x\right) = \rho_{out}(g) P(x) \rho_{in}(g)^{-1} \quad \forall g \in G, x \in \mathbb{R}^d. \quad (4)$$

$P(x)$ just means that $x$ is plugged into the polynomials in each entry of $P$, which results in a real-valued matrix $P(x) \in \mathbb{R}^{c_{out} \times c_{in}}$, and similarly for $P\left((g^{-1})^T x\right)$. Appendix C provides more intuition for the action of group elements on polynomials. Because $D$ is an isomorphism, this result completely characterizes when a PDO with constant coefficients is $G$-equivariant, and in conjunction with Proposition 1, when any PDO between feature fields is $H$-equivariant. The proof of Theorem 2 can again be found in Appendix G.

To avoid confusion, we would like to point out that this description of PDOs as polynomials is only a useful trick that lets us express certain operations more easily and will later let us connect steerable PDOs to steerable kernels. Convolving with these polynomials is *not* a meaningful operation; they only become useful when $\partial$ is plugged in for $x$.

## 2.3 EXAMPLES OF EQUIVARIANT PDOS

To build intuition, we explore the space of equivariant PDOs for two simple cases before covering the general solution. Consider $\mathbb{R}^2$ as a base space with the symmetry group $G = \mathrm{SO}(2)$ and trivial representations $\rho_{\mathrm{in}}$ and $\rho_{\mathrm{out}}$, i.e. maps between two scalar fields. The equivariance condition then becomes $p(gx) = p(x)$, so the polynomial $p$ has to be rotation invariant. This is the case if and only if it can be written as a function of $|x|^2$, i.e. $p(x) = q(|x|^2)$ for some $q : \mathbb{R}_{\geq 0} \to \mathbb{R}$. Since we want $p$ to be a polynomial, $q$ needs to be a polynomial in one variable. Because $|x|^2 = x_1^2 + x_2^2$, we get the PDO $D(p) = q(\Delta)$, where $\Delta := \partial_1^2 + \partial_2^2$ is the Laplace operator. So the $\mathrm{SO}(2)$-equivariant PDOs between two scalar fields are exactly the polynomials in the Laplacian, such as $2\Delta^2 + \Delta + 3$.

As a second example, consider PDOs that map from a vector to a scalar field, still with $\mathrm{SO}(2)$ as the symmetry group. There are two such PDOs that often occur in the natural sciences, namely the divergence, $\mathrm{div}\, v := \partial_1 v_1 + \partial_2 v_2$, and the 2D curl, $\mathrm{curl}_{2D}\, v := \partial_1 v_2 - \partial_2 v_1$. Both of these are $\mathrm{SO}(2)$-equivariant (see Appendix A). We get additional equivariant PDOs by composing with equivariant scalar-to-scalar maps, i.e. polynomials in the Laplacian. Specifically, $q(\Delta) \circ \mathrm{div}$ and $q(\Delta) \circ \mathrm{curl}_{2D}$ are also equivariant vector-to-scalar PDOs, for any polynomial $q$. We will omit the $\circ$ from now on.

We show in Appendix A that these PDOs already span the *complete* space of $\mathrm{SO}(2)$-equivariant PDOs from vector to scalar fields. Explicitly, the equivariant PDOs in this setting are all of the form $q_1(\Delta)\, \mathrm{div} + q_2(\Delta)\, \mathrm{curl}_{2D}$ for polynomials $q_1$ and $q_2$. One example of this is the PDO shown in Fig. 1.

In these examples, as well as in other simple cases (see Appendix A), the equivariant PDOs are all combinations of well-known operators such as the divergence and Laplacian. That the notion of equivariance can reproduce all the intuitively "natural" differential operators suggests that it captures the right concept.

## 3 BASES FOR SPACES OF STEERABLE PDOS

For the purposes of deep learning, we need to be able to learn steerable PDOs. To illustrate how to achieve this, consider again $\mathrm{SO}(2)$-equivariant PDOs mapping from vector to scalar field. We have seen in Section 2.3 that they all have the form

$$q_1(\Delta)\operatorname{div} + q_2(\Delta)\operatorname{curl}_{2\mathrm{D}} . \tag{5}$$

In practice, we will need to limit the order of the PDOs we consider, so that we can discretize them. For example, we could consider only polynomials $q_1$ and $q_2$ of up to first order, i.e. $q_1(z) = c_1 z + c_2$ and $q_2(z) = c_3 z + c_4$. This leads to PDOs of up to order three (since $\Delta$ is a second order PDO and div and $\operatorname{curl}_{2\mathrm{D}}$ are first order). The space of such equivariant PDOs is then

$$c_1\Delta\operatorname{div} + c_2\operatorname{div} + c_3\Delta\operatorname{curl}_{2\mathrm{D}} + c_4\operatorname{curl}_{2\mathrm{D}} , \qquad c_i \in \mathbb{R} . \tag{6}$$

We can now train the real-valued parameters $c_1, \ldots, c_4$ and thereby learn arbitrary equivariant PDOs of up to order three.

The general principle is that we need to find a *basis* of the real vector space of steerable PDOs; then we learn weights for a linear combination of the basis elements, yielding arbitrary equivariant PDOs.

Different group representations are popular in equivariant deep learning practice; for example PDO-eConvs and group convolutions correspond to so-called *regular* representations (see Appendix M) but quotient representations have also been used very successfully (Weiler & Cesa, 2019). We therefore want to find bases of steerable PDOs for *arbitrary* representations $\rho_{\mathrm{in}}$ and $\rho_{\mathrm{out}}$. To do so, we make use of the existing work on solving the closely related $G$-steerability constraint for *kernels*. In this section, we will first give a brief overview of this kernel steerability constraint and then describe how to transfer its solutions to ones of the *PDO* steerability constraint.

### 3.1 THE STEERABILITY CONSTRAINT FOR KERNELS

The kernel $G$-steerability constraint characterizes when convolution with a kernel $\kappa : \mathbb{R}^d \to \mathbb{R}^{c_{\mathrm{out}} \times c_{\mathrm{in}}}$ is $G$-equivariant, like the PDO steerability constraint Eq. (4) does for PDOs. Namely,

$$\kappa(gx) = |\det g|^{-1} \rho_{\mathrm{out}}(g)\kappa(x)\rho_{\mathrm{in}}(g)^{-1} \qquad \forall g \in G \tag{7}$$

has to hold. This constraint was proven in (Weiler et al., 2018a) for orthogonal $G$ and later in (Weiler et al., 2021) for general $G$. It is very similar to the PDO steerability constraint: the only differences are the determinant and the $\kappa(gx)$ term on the LHS where we had $(g^{-1})^T$ instead of $g$.

An explanation for this similarity is that both constraints can be seen as special cases of a more general steerability constraint for *Schwartz distributions*, which we derive in Appendix E. Schwartz distributions are a generalization of classical functions and convolutions with Schwartz distributions can represent both classical convolutions and PDOs. As we prove in Appendix E, such distributional convolutions are the most general translation equivariant, continuous, linear maps between feature spaces, strictly more general than either PDOs or classical kernels. We also show how steerable PDOs can be interpreted as the Fourier transform of steerable kernels, which we use to explain the remaining differences between the two steerability constraints.

But for the purposes of this section, we want to draw particular attention to the fact that for $G \le \mathrm{O}(d)$, the two constraints become exactly identical: the determinant then becomes 1, and $(g^{-1})^T = g$. So for this case, which is by far the most practically important one, we will use existing solutions of the kernel steerability constraint to find a complete basis of equivariant PDOs.

### 3.2 TRANSFERRING KERNEL SOLUTIONS TO STEERABLE PDO BASES

Solutions of the kernel steerability constraint have been published for subgroups of $\mathrm{O}(2)$ (Lang & Weiler, 2021; Weiler & Cesa, 2019) and $\mathrm{O}(3)$ (Lang & Weiler, 2021; Weiler et al., 2018a), and they

all use a basis of the form

$$\{\kappa_{\alpha\beta}(x) := \varphi_\alpha(|x|)\chi_\beta(x/|x|) \,|\, \alpha \in \mathcal{A}, \beta \in \mathcal{B}\}\,, \tag{8}$$

where $\mathcal{A}$ and $\mathcal{B}$ are index sets for the radial and angular part and the $\varphi_\alpha$ span the entire space of radial functions. The reason is that the steerability constraint Eq. (7) constrains only the angular part if $G \le \mathrm{O}(d)$, because orthogonal groups preserve distances.

The angular functions $\chi_\beta$ are only defined on the sphere $S^{d-1}$. But crucially, we show in Appendix J that they can all be canonically extended to polynomials defined on $\mathbb{R}^d$.[3] Concretely, we define $\tilde{\chi}_\beta(x) := |x|^{l_\beta}\chi_\beta(x/|x|)$, where $l_\beta$ is chosen minimally such that $\tilde{\chi}_\beta$ is a polynomial. What we prove in Appendix J is that such an $l_\beta$ always exists.

The radial part $\varphi_\alpha$ in the kernel basis consists of *unrestricted* radial functions. To get a basis for the space of *polynomial* steerable kernels, it is enough to use only powers of $|x|^2$ for the radial part. Specifically, we show in Appendix H that

$$\left\{ p_{k\beta} := |x|^{2k}\tilde{\chi}_\beta \,\middle|\, k \in \mathbb{N}_0, \beta \in \mathcal{B} \right\} \tag{9}$$

is a basis for the space of polynomial steerable kernels. As a final step, we interpret each polynomial as a PDO using the isomorphism $D$ defined in Section 2.2, yielding a *complete* basis of the space of steerable PDOs. The $|x|^{2k}$ terms become Laplacian powers $\Delta^k$, while in the examples from Section 2.3, $\tilde{\chi}_\beta$ corresponds to divergence and curl.

In Appendix I, we apply this procedure to subgroups of $\mathrm{O}(2)$ and $\mathrm{O}(3)$ to obtain concrete solutions.

## 4 EXPERIMENTS

**Implementation**   We developed the theory of steerable PDOs in a continuous setting, but for implementation the PDOs need to be discretized, just like steerable kernels. The method of discretization is completely independent of the steerable PDO basis, so steerable PDOs can be combined with any discretization procedure. We compare three methods, *finite differences* (FD), *radial basis function finite differences* (RBF-FD) and *Gaussian derivatives*.

Finite differences are a generalization of the usual central difference approximation and are the method used by PDO-eConvs (Shen et al., 2020). RBF-FD finds stencils by demanding that the discretization should become exact when applied to radial basis functions placed on the stencil points. Its advantage over FD is that it can be applied to structureless point clouds rather than only to regular grids. Gaussian derivative stencils work by placing a Gaussian on the target point and then evaluating its derivative on the stencil points. Like RBF-FD, this also works on point clouds, and the Gaussian also has a slight smoothing effect, which is why this discretization is often used in Computer Vision. Formal descriptions of all three discretization methods can be found in Appendix L.

In addition to discretization, the infinite basis of steerable PDOs or kernels needs to be restricted to a finite subspace. For kernels, we use the bandlimiting filters by Weiler & Cesa (2019). For PDOs, we limit the total derivative order to two for $3 \times 3$ stencils and to three for $5 \times 5$ stencils (except for PDO-eConvs, where we use the original basis that limits the maximum order of *partial* derivatives).

Finally, steerable PDOs and steerable kernels only replace the convolutional layers in a classical CNN. To achieve an equivariant network, all the other layers, such as nonlinearities or Batchnorm also need to be equivariant. Weiler & Cesa (2019) discuss in details how this can be achieved for various types of layers. In our experiments, we use exactly the same implementation they do. Care also needs to be taken with biases in the PDO layers. Here, we again follow (Weiler & Cesa, 2019) by adding a bias only to the trivial irreducible representations that make up $\rho_{\mathrm{out}}$.

---

[3]This holds for the cases we discuss here, i.e. subgroups of $\mathrm{O}(2)$ and $\mathrm{O}(3)$. In higher dimensions, the situation is more complicated, but for those the kernel solutions have not yet been worked out anyway.

[4]As in the original PDO-eConv paper (Shen et al., 2020). Note that their performance is better, which is simply caused by their different architecture and hyperparameters.

**Rotated MNIST** We first benchmark steerable PDOs on rotated MNIST (Larochelle et al., 2007), which consists of MNIST images that have been rotated by different angles, with 12k train and 50k test images. Our results can be found in Table 1. The models with $5 \times 5$ stencils use an architecture that Weiler & Cesa (2019) used for steerable CNNs, with six $C_{16}$-equivariant layers followed by two fully connected layers. The first column gives the representation under which the six equivariant layers transform (see Appendix B for their definitions). PDO-eConvs implicitly use regular representations (see Appendix M), but with a slightly different basis than the one we present, so we test both bases. We also tested models that are $D_{16}$-equivariant in their first layers and $C_{16}$-equivariant in their last one but did not find any improvements, see Appendix N. For the models with $3 \times 3$ stencils, we use eight instead of six $C_{16}$-equivariant layers, in order to compensate for the smaller receptive field and keep the parameter count comparable. The remaining differences between kernel and PDO parameter counts come from the fact that the basis restrictions necessarily work slightly differently (via bandlimiting filters or derivative order restriction respectively). All models were trained with 30 epochs and hyperparameters based on those by Weiler & Cesa (2019), though we changed the learning rate schedule and regularization slightly because this improved performance for all models, including kernel-based ones. The training data is augmented with random rotations. Precise descriptions of the architecture and hyperparameters can be found in Appendix O.

Table 1: MNIST-rot results. Test errors $\pm$ standard deviations are averaged over six runs. Vanilla CNN is a solely translation equivariant model ($G = \{e\}$) with the same general architecture. See main text for details on the models.

| Representation | Method | Stencil | Error [%] | Params |
|---|---|---|---|---|
| – | Vanilla CNN | $3 \times 3$ 
 $5 \times 5$ | $2.001 \pm 0.030$ 
 $1.959 \pm 0.055$ | 1.1M |
| regular (our basis) | Kernels | $3 \times 3$ 
 $5 \times 5$ | $0.741 \pm 0.036$ 
 $0.683 \pm 0.021$ | 837K 
 1.1M |
| | FD | $3 \times 3$ 
 $5 \times 5$ | $1.196 \pm 0.062$ 
 $1.54 \ \pm 0.32$ | 837K 
 941K |
| | RBF-FD | $3 \times 3$ 
 $5 \times 5$ | $1.313 \pm 0.065$ 
 $1.475 \pm 0.020$ | 837K 
 941K |
| | Gauss | $3 \times 3$ 
 $5 \times 5$ | $0.795 \pm 0.030$ 
 $0.750 \pm 0.017$ | 837K 
 941K |
| regular (PDO-eConv) | FD$^4$ | $5 \times 5$ | $1.98 \ \pm 0.11$ | 982K |
| | Gauss | $5 \times 5$ | $0.831 \pm 0.039$ | |
| quotient (our basis) | Kernels | $3 \times 3$ 
 $5 \times 5$ | $0.717 \pm 0.026$ 
 $0.670 \pm 0.011$ | 877K 
 1.1M |
| | FD | $3 \times 3$ 
 $5 \times 5$ | $1.143 \pm 0.063$ 
 $1.347 \pm 0.026$ | 877K 
 951K |
| | RBF-FD | $3 \times 3$ 
 $5 \times 5$ | $1.303 \pm 0.077$ 
 $1.422 \pm 0.040$ | 877K 
 951K |
| | Gauss | $3 \times 3$ 
 $5 \times 5$ | $0.825 \pm 0.053$ 
 $0.744 \pm 0.040$ | 877K 
 951K |

**STL-10** The rotated MNIST dataset has global rotational symmetry by design, so it is unsurprising that equivariant models perform well. But interestingly, rotation equivariance can also help for natural images without global rotational symmetry (Weiler & Cesa, 2019; Shen et al., 2020). We therefore benchmark steerable PDOs on STL-10 (Coates et al., 2011), where we only use the labeled portion of 5000 training images. The results are shown in Table 2. The model architecture and hyperparameters are exactly the same as in (Weiler & Cesa, 2019), namely a Wide-ResNet-16-8 trained for 1000 epochs with random crops, horizontal flips and Cutout (De-Vries & Taylor, 2017) as data augmentation. The group column describes the equivariance group in each of the three residual blocks. For example, $D_8 D_4 D_1$ means that the first block is equivariant under reflections and 8 rotations, the second under 4 rotations and the last one only under reflections. All layers use regular representations. The $D_8$-equivariant layers use $5 \times 5$ filters to improve equivariance, whereas the other layers use $3 \times 3$ filters.

Table 2: STL-10 results, again over six runs. All models except the vanilla CNN use regular representations, see main text for details.

| Method | Groups | Error [%] | Params |
|---|---|---|---|
| Vanilla CNN | – | $12.7 \pm 0.2$ | 11M |
| Kernels | $D_8 D_4 D_1$ 
 $D_4 D_4 D_1$ | $10.7 \pm 0.6$ 
 $10.2 \pm 0.4$ | 4.2M |
| FD | $D_8 D_4 D_1$ 
 $D_4 D_4 D_1$ | $12.1 \pm 0.6$ 
 $12.1 \pm 0.7$ | |
| RBF-FD | $D_8 D_4 D_1$ 
 $D_4 D_4 D_1$ | $14.3 \pm 0.4$ 
 $14.3 \pm 0.4$ | 3.2M |
| Gauss | $D_8 D_4 D_1$ 
 $D_4 D_4 D_1$ | $11.2 \pm 0.3$ 
 $10.6 \pm 0.8$ | |

**Fluid flow prediction**    In the previously described tasks, the input and output representations are all trivial. To showcase the use of non-trivial output representations, we predict laminar fluid flow around various objects, following Ribeiro et al. (2020). In this case, the network outputs a *vector field*, which behaves differently under rotations than scalar outputs, and whose equivariance cannot be represented using PDO-eConvs, since they only implement trivial and regular representations. We find that equivariance significantly improves performance for both kernels and PDOs, compared to vanilla non-equivariant versions. See Appendix N for details.

**Equivariance errors**    In the continuum, steerable CNNs and steerable PDOs are both *exactly* equivariant. But the discretization on a square grid leads to unavoidable equivariance errors for rotations that aren't multiples of $\frac{\pi}{2}$. The violation of equivariance in practice is thus closely connected to the discretization error. For finite differences, the discretization error is particularly easy to bound asymptotically, and as pointed out by Shen et al. (2020), this places the same asymptotic bound on the equivariance error. However, our experiments show that empirically, finite differences don't lead to a particularly low equivariance error (kernels and all PDO discretizations perform similarly). See Table 4 in Appendix N for details.

**Locality of PDOs**    While all equivariant models improve significantly over the non-equivariant CNN, the method of discretization plays an important role for PDOs. The reason that FD and RBF-FD underperform kernels is that they don't make full use of the stencil, since PDOs are inherently local operators. When a $5 \times 5$ stencil is used, the outermost entries are all very small compared to the inner ones, and even in $3 \times 3$ kernels, the four corners tend to be closer to zero (see Appendix N for images of stencils to illustrate this). Gaussian discretization performs significantly better and almost as well as kernels because its smoothing effect alleviates these issues. This fits the observation that kernels and Gaussian methods profit from using $5 \times 5$ kernels, whereas these do not help for FD and RBF-FD (and in fact decrease performance because of the smaller number of layers).

## 5    CONCLUSION

We have described a general framework for equivariant PDOs acting on feature fields over Euclidean space. With this framework, we found strong similarities between equivariant PDOs and equivariant convolutions, even unifying the two using convolutions with Schwartz distributions. We exploited these similarities to find bases for equivariant PDOs based on existing solutions for steerable kernels.

Our experiments show that the locality of PDOs can be a disadvantage compared to convolutional kernels. However, our approach for equivariance can easily be combined with any discretization method, and we show that using Gaussian derivatives for discretization alleviates the issue. Equivariant PDOs could also be very useful in cases where their strong locality is a desideratum rather than a drawback.

The theory developed in this work provides the necessary foundation for applications where equivariant PDOs, rather than kernels, are needed. For example, Probabilistic Numerical CNNs (Finzi et al., 2021) use PDOs in order to parameterize convolutions on *continuous* input data. Finzi et al. also derive a constraint to make these PDOs equivariant, which is a special case of our PDO $G$-steerability constraint Eq. (4). The solutions to this constraint presented in this paper are the missing piece for implementing and empirically evaluating Probabilistic Numerical CNNs – neither of which Finzi et al. do.

Another promising application of PDOs is an extension to manifolds. Gauge CNNs (Cohen et al., 2019b; Kicanaoglu et al., 2019; Haan et al., 2021; Weiler et al., 2021) are a rather general framework for convolutions on manifolds; see Weiler et al. (2021) for a thorough treatment and literature review. As in the Euclidean case, Gauge CNNs use feature fields that transform according to some representation $\rho$. The kernels are defined on the tangent space and are still constrained by the $G$-steerability constraint. Because of that, our approach to equivariant Euclidean PDOs is very well-suited for generalization to manifolds and our steerable PDO solutions will still remain valid. One advantage of PDOs in this setting is that they require no Riemannian structure and can achieve equivariance with respect to arbitrary diffeomorphisms, as is common in physics, instead of mere isometry equivariance.

## REPRODUCIBILITY STATEMENT

All Propositions and Theorems in the main paper and in the appendix explicitly state all their assumptions. In cases where theoretical results are stated only informally in the main paper (such as our results involving Schwartz distributions), the precise claims can be found in the appendix. The appendix also contains complete proofs for all of our theoretical claims.

The code necessary to reproduce our experiments can be found at `https://github.com/ejnnr/steerable_pdo_experiments`. The required datasets are downloaded and preprocessed automatically. The exact hyperparameters we used are available as pre-defined configuration options in our scripts. We also include a version lockfile for installing precisely the right versions of all required Python packages. Our implementation of steerable PDOs is easy to adapt to different use cases and fully documented, allowing other practitioners to test the method on different datasets or tasks.

## ACKNOWLEDGMENTS

We would like to thank Gabriele Cesa and Leon Lang for discussions on integrating steerable PDOs into the E2CNN library and on solutions of the kernel $G$-steerability constraint. This work was supported by funding from QUVA Lab.

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

SUPPLEMENTARY MATERIAL

## A   STEERABLE PDOs BETWEEN VECTOR AND SCALAR FIELDS

We give bases for the space of steerable PDOs for many important cases in Appendix K. However, the generality of the description there obscures the connection to well-known PDOs such as the gradient or divergence. So to complement the general solutions, we discuss a few simple cases in much more detail in this section. We will see that Laplacian, gradient, divergence, and curl are all rotation equivariant and, more interestingly, that all rotation equivariant PDOs can be constructed by combining these (in the simple settings we cover in this section). In many cases, we rederive the solutions even though all of them would follow immediately from the general case in Appendix K, in order to provide some intuition on why these are the only equivariant PDOs. Readers who are only interested in an overview of the results may wish to skip to the end of this section.

### A.1   SCALAR TO SCALAR PDOs

We have already argued in Section 2.3 that the $\mathrm{SO}(2)$-equivariant PDOs between two scalar fields are precisely polynomials in the Laplacian, i.e. of the form $q(\Delta)$ for an arbitrary real polynomial $q \in \mathbb{R}[x]$. The derivation given there applies without changes to $\mathrm{SO}(d)$ for $d > 2$ and to $\mathrm{O}(d)$ as well, so the same holds in these cases.

### A.2   SCALAR TO VECTOR PDOs

We start by considering the case where $\rho_{\mathrm{in}}$ is trivial (with $c_{\mathrm{in}} = 1$) and $\rho_{\mathrm{out}}$ is the vector field representation (i.e. $c_{\mathrm{out}} = d$). We can then represent PDOs by a $d \times 1$ matrix of polynomials, i.e. a column vector. The PDO steerability constraint becomes

$$P(gx) = gP(x) \tag{10}$$

since $\rho_{\mathrm{in}}$ is trivial. A good mental model for this subsection is to think of $P$ as a vector field $P : \mathbb{R}^d \to \mathbb{R}^d$ whose entries happen to be polynomials. The steerability constraint simply states that this vector field must "look the same" after a rotation, see Fig. 3 for examples.

We begin by discussing the case $G = \mathrm{SO}(2)$, which is somewhat different from $\mathrm{O}(2)$ and from $d > 2$. Any rotation equivariant vector field on $\mathbb{R}^2$ is fully determined by its values on the ray $\{(x_1, 0) \,|\, x_1 > 0\}$. Specifically, the rotation equivariant vector fields $v$ are precisely those that can in polar coordinates be written as

$$v(r, \varphi) = \begin{pmatrix} \cos\varphi & -\sin\varphi \\ \sin\varphi & \cos\varphi \end{pmatrix} \begin{pmatrix} f_1(r) \\ f_2(r) \end{pmatrix} \tag{11}$$

for arbitrary radial parts $f_1, f_2$. In Cartesian coordinates, we can write this as

$$v(x, y) = \begin{pmatrix} x & -y \\ y & x \end{pmatrix} \begin{pmatrix} \tilde{f}_1(x^2 + y^2) \\ \tilde{f}_2(x^2 + y^2) \end{pmatrix} \tag{12}$$

where $\tilde{f}_i(z) := \frac{f_i(\sqrt{z})}{\sqrt{z}}$. We need $v_1$ and $v_2$ to be polynomials in $x, y$, which means that the $\tilde{f}_i$ need to be polynomials. If we then apply the $D$ isomorphism, we get:

**Proposition 3.** *The* $\mathrm{SO}(2)$*-equivariant differential operators from a scalar to a vector field are exactly those of the form*

$$\begin{pmatrix} \partial_1 & -\partial_2 \\ \partial_2 & \partial_1 \end{pmatrix} \begin{pmatrix} q_1(\Delta) \\ q_2(\Delta) \end{pmatrix} = q_1(\Delta) \begin{pmatrix} \partial_1 \\ \partial_2 \end{pmatrix} + q_2(\Delta) \begin{pmatrix} -\partial_2 \\ \partial_1 \end{pmatrix} \tag{13}$$

*where $q_1$ and $q_2$ are arbitrary polynomials.*

In words, the $\mathrm{SO}(2)$-equivariant PDOs are all linear combinations of the gradient and the transpose of the 2D curl, $(-\partial_2, \partial_1)^T$, with coefficients being polynomials in the Laplacian (rather than just real numbers).

---

[5]Created using `https://www.desmos.com/calculator/eijhparfmd`

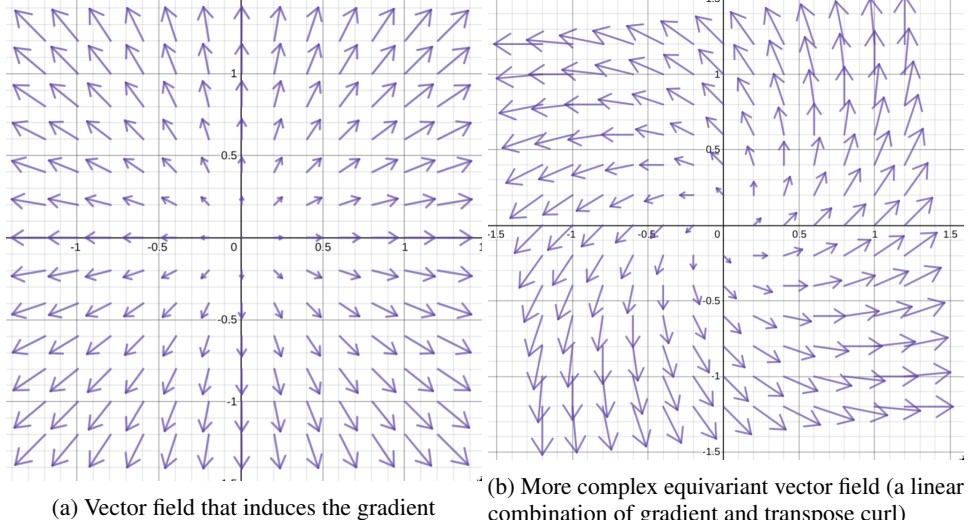

(a) Vector field that induces the gradient

(b) More complex equivariant vector field (a linear combination of gradient and transpose curl)

Figure 3: Two examples of SO(2)-equivariant vector fields.[5]

The gradient is also equivariant under reflections, i.e. O(2)-equivariant, and it easily generalizes to higher dimensions. However, the transpose 2D curl only appears in this particular setting: it is not reflection-equivariant, and it does not have an analogon in higher dimensions.[6] We summarize this in the following result:

**Proposition 4.** *Let $G = \mathrm{O}(d)$ for $d \geq 2$ or $G = \mathrm{SO}(d)$ for $d > 2$. Then the G-equivariant differential operators from a scalar to a vector field are exactly those of the form*

$$q(\Delta)\,\mathrm{grad} \tag{14}$$

*for an arbitrary polynomial $q \in \mathbb{R}[x]$.*

A possible intuition for why SO(2) is a special case is that $\mathrm{SO}(2) \cong S^1$ whereas $\mathrm{SO}(d) \not\cong S^{d-1}$ for $d > 2$ and $\mathrm{O}(d) \not\cong S^{d-1}$ for $d \geq 2$. Our construction above heavily makes use of the fact that rotations of $\mathbb{R}^2$ correspond one-to-one to angles, i.e. points on $S^1$, and this construction thus does not generalize to any other cases.

Before we prove Proposition 4, we show a helpful lemma:

**Lemma 5.** *Let $d > 2$. Then for any linearly independent vectors $v, w \in \mathbb{R}^d$, there is a rotation $g \in \mathrm{SO}(d)$ such that $gv = v$ but $gw \neq w$. For $d = 2$, there is such a $g \in \mathrm{O}(2)$.*

*Proof of lemma.* Since $v$ and $w$ are linearly independent, we can write $\mathbb{R}^d = \mathrm{span}(v) \oplus W$, where $W$ is a linear subspace containing $w$. Pick an element $\tilde{g} \in \mathrm{SO}(d-1)$ (or $\mathrm{O}(d-1) = \mathrm{O}(1) = \{\pm 1\}$ in the $d = 2$ case) such that $\tilde{g}w \neq w$, where $\tilde{g}$ acts on $W$ (this always exists, note that $w \neq 0$). Then there is a $g \in \mathrm{SO}(d)$ (or $\mathrm{O}(d)$) that restricts to $\tilde{g}$ on $W$ and is the identity on $\mathrm{span}(v)$ (in terms of matrices with respect to a basis of the form $\{v, \dots\}$, $g$ would be block-diagonal, with a $1 \times 1$ identity block and a $(d-1) \times (d-1)$ block for $\tilde{g}$). $\qquad\square$

*Proof of Proposition 4.* One direction is clear: the gradient is induced by the matrix of polynomials $P(x) = x$, $x \in \mathbb{R}^d$, which is clearly equivariant. We have also seen that polynomials of the Laplacian are equivariant, and we can compose these equivariant PDOs to get another equivariant PDO. So what remains to show is that no other equivariant PDOs exist.

---

[6]To avoid confusion, we remark that the 3D curl of course exists but maps between two vector fields. The 2D curl is in fact closely related to the 3D curl, and the fact that it does not have a higher-dimensional analogon corresponds to the fact that the 3D curl, as a vector to vector PDO cannot easily be generalized to higher dimensions.

So let $P$ be $G$-equivariant, with $G$ as in Proposition 4. Let furthermore $x \in \mathbb{R}^d$ be arbitrary and $g \in G$ be an element in the stabilizer of $x$, i.e. $gx = x$. Then we have

$$gP(x) = P(gx) = P(x). \tag{15}$$

In other words, for any $g \in G$ with $gx = x$, we also have $gP(x) = P(x)$. By the lemma, $x$ and $P(x)$ are thus linearly dependent, i.e. $P(x) = cx$ for this particular $x$ and some $c \in \mathbb{R}$.

We can apply this reasoning to any $x \in \mathbb{R}^d$, so there is a function $c : \mathbb{R}^d \to \mathbb{R}$ such that $P(x) = c(x)x$. Furthermore,

$$c(x)gx = gP(x) = P(gx) = c(gx)gx, \tag{16}$$

so $c$ has to be rotation invariant. As we have already argued, this implies that $c(x) = q(x_1^2 + \ldots + x_d^2)$ for some function $q$, and since $c$ needs to be a polynomial, so does $q$. Then

$$D(P) = q(\Delta) \operatorname{grad} \tag{17}$$

as claimed. $\qquad\square$

If $d = 1$, the lemma also holds and the proof goes through – this is just the scalar case from the previous section, which is generalized here. But for $G = \mathrm{SO}(2)$, this argument does not work because $SO(1) = \{1\}$, so the decisive step in the proof of the lemma fails. That is what allows the additional equivariant PDOs.

## A.3   VECTOR TO SCALAR PDOS

Equivariant PDOs mapping from vector to scalar fields are simply the transpose of those mapping from scalar to vector fields (for orthogonal groups $G$): $P$ is now a $1 \times d$ matrix and the steerability constraint is

$$P(gx) = P(x)g^{-1} = P(x)g^T. \tag{18}$$

By transposing, we get

$$P^T(gx) = gP^T(x), \tag{19}$$

which is the equivariance condition for scalar to vector layers. As solutions, we get the divergence (as the transpose of the gradient) and for $G = \mathrm{SO}(2)$ also the 2D curl. They are again combined linearly with polynomials in the Laplacian as coefficients.

## A.4   VECTOR TO VECTOR PDOS

For PDOs mapping between two vector fields, the steerability constraint is

$$P(gx) = gP(x)g^{-1}. \tag{20}$$

Since the solution space in this case is somewhat more complicated, we will only cover $G = \mathrm{SO}(2)$ in these examples; see Appendix K for more solutions. In principle, we could apply the same method that we already used before for $\mathrm{SO}(2)$, choosing the radial components of $P$ freely and using the steerability constraint to determine the angular components. But since the computations in this case are more involved and don't yield much additional insight, we will instead use the solutions from Appendix K. Vector fields correspond to frequency 1 irreps, and writing out the solutions for those explicitly, we get that the equivariant PDOs are exactly linear combinations of

$$\begin{pmatrix} 1 & 0 \\ 0 & 1 \end{pmatrix}, \quad \begin{pmatrix} 0 & -1 \\ 1 & 0 \end{pmatrix}, \quad \begin{pmatrix} \partial_x^2 - \partial_y^2 & 2\partial_x\partial_y \\ 2\partial_x\partial_y & \partial_y^2 - \partial_x^2 \end{pmatrix}, \quad \begin{pmatrix} -2\partial_x\partial_y & \partial_x^2 - \partial_y^2 \\ \partial_x^2 - \partial_y^2 & 2\partial_x\partial_y \end{pmatrix} \tag{21}$$

with polynomials in the Laplacian as coefficients.

The first two operators are simply the identity and a $\frac{\pi}{2}$ rotation matrix, both zeroth order PDOs. Note that the rotation matrix rotates the fibers of the vector field, it does not act on the base space. The other two operators are less interpretable, but we can replace them through a change of basis:

$$\frac{1}{2}\Delta \begin{pmatrix} 1 & 0 \\ 0 & 1 \end{pmatrix} + \frac{1}{2} \begin{pmatrix} \partial_x^2 - \partial_y^2 & 2\partial_x\partial_y \\ 2\partial_x\partial_y & \partial_y^2 - \partial_x^2 \end{pmatrix} = \begin{pmatrix} \partial_x^2 & \partial_x\partial_y \\ \partial_x\partial_y & \partial_y^2 \end{pmatrix}, \tag{22}$$

which is the matrix describing the composition grad $\circ$ div. Similarly,

$$-\frac{1}{2}\Delta\begin{pmatrix} 0 & -1 \\ 1 & 0 \end{pmatrix} + \frac{1}{2}\begin{pmatrix} -2\partial_x\partial_y & \partial_x^2 - \partial_y^2 \\ \partial_x^2 - \partial_y^2 & 2\partial_x\partial_y \end{pmatrix} = \begin{pmatrix} -\partial_x\partial_y & \partial_x^2 \\ -\partial_y^2 & \partial_x\partial_y \end{pmatrix}, \tag{23}$$

which is the matrix describing grad $\circ$ curl$_{\mathrm{2D}}$. So if we write $R$ for the $\frac{\pi}{2}$ rotation matrix (interpreted as a PDO), then the equivariant PDOs mapping between vector fields are exactly linear combinations of

$$\mathrm{id}, \quad R, \quad \mathrm{grad} \circ \mathrm{div}, \quad \mathrm{grad} \circ \mathrm{curl}_{\mathrm{2D}}, \tag{24}$$

as always with polynomials in the Laplacian as coefficients.

We can be even more economical and describe these PDOs with fewer building blocks. For two vectors $P, Q$ of one-dimensional PDOs, e.g. $P = \partial = (\partial_1, \partial_2)^T$, we write $P \otimes Q$ for the $2 \times 2$ PDO with entries $(P \otimes Q)_{ij} = P_i Q_j$. Then we can write for example grad $\circ$ div $= \partial \otimes \partial$. We furthermore note that

$$R\partial = \begin{pmatrix} 0 & -1 \\ 1 & 0 \end{pmatrix}\begin{pmatrix} \partial_x \\ \partial_y \end{pmatrix} = \begin{pmatrix} -\partial_y \\ \partial_x \end{pmatrix} = \mathrm{curl}_{\mathrm{2D}} . \tag{25}$$

This means we can write the basis from above as

$$\mathrm{id}, \quad R, \quad \partial \otimes \partial, \quad \partial \otimes R\partial . \tag{26}$$

## A.5 SUMMARY OF RESULTS

- $\mathrm{SO}(d)$- or $\mathrm{O}(d)$-equivariant PDOs between two scalar fields are exactly polynomials in the Laplacian.

- $G$-equivariant PDOs mapping from scalar to vector fields are exactly those of the form $q(\Delta)\,\mathrm{grad}$ for $G = \mathrm{SO}(d)$ with $d > 2$ or $G = \mathrm{O}(d)$. For PDOs from vector to scalar fields, we similarly get $q(\Delta)\,\mathrm{div}$.

- $\mathrm{SO}(2)$-equivariant PDOs from vector to scalar fields are exactly those of the form

$$q_1(\Delta)\,\mathrm{div} + q_2(\Delta)\,\mathrm{curl}_{\mathrm{2D}} . \tag{27}$$

  For PDOs from scalar to vector fields, we get grad instead of div and the transpose of the 2D curl instead.

- $\mathrm{SO}(2)$-equivariant PDOs between two vector fields are linear combinations of the identity, a $\frac{\pi}{2}$ rotation, grad $\circ$ div and grad $\circ$ curl$_{\mathrm{2D}}$, with polynomials in the Laplacian as coefficients.

## B REPRESENTATION THEORY PRIMER

This section introduces the fundamental definitions of representation theory that we need.

### B.1 BASIC DEFINITIONS

**Definition 1.** A *group representation* of a group $G$, or *representation* for short, is a group homomorphism $\rho : G \to \mathrm{GL}(V)$ for some vector space $V$. This means that $\rho(gg') = \rho(g)\rho(g')$ for all $g, g'$, so multiplication of group elements in $G$ is represented as matrix multiplication in $\mathrm{GL}(V)$.

For this paper, we only need $V = \mathbb{R}^c$, so we focus on this case. In particular, some of the following definitions make use of the fact that we only consider finite-dimensional representations.

Given multiple representations of the same group, we can "stack them together" using direct sums:

**Definition 2.** Let $\rho_1 : G \to \mathrm{GL}(\mathbb{R}^{c_1})$ and $\rho_2 : G \to \mathrm{GL}(\mathbb{R}^{c_2})$ be two representations of $G$. Then we define the *direct sum* representation $\rho_1 \oplus \rho_2 : G \to \mathrm{GL}(\mathbb{R}^{c_1+c_2})$ by

$$(\rho_1 \oplus \rho_2)(g) := \begin{pmatrix} \rho_1(g) & \\ & \rho_2(g) \end{pmatrix}, \tag{28}$$

which acts independently on the subspaces $\mathbb{R}^{c_1}$ and $\mathbb{R}^{c_2}$ of $\mathbb{R}^{c_1+c_2}$.

This corresponds exactly to stacking feature fields of different types. For example, we can stack a vector and a scalar field into one four-dimensional field, such that its vector and scalar part transform *independently*. The representation of this four-dimensional field will be the direct sum of the vector and scalar field representations.

Often, two representations are formally different but can be transformed into one another using a change of basis; they then behave the same in all relevant aspects. For example, $SO(2)$ has a representation

$$\rho(\varphi) := \begin{pmatrix} \cos \varphi & -\sin \varphi \\ \sin \varphi & \cos \varphi \end{pmatrix} \tag{29}$$

that represents a rotation angle $\varphi$ by a counterclockwise rotation matrix (this representation is the one used for vector fields). However, we could just as well use

$$\rho(\varphi) := \begin{pmatrix} \cos \varphi & \sin \varphi \\ -\sin \varphi & \cos \varphi \end{pmatrix} , \tag{30}$$

where the rotation is clockwise. Using one or the other is pure convention and we would like to treat them as "the same" representation. This is formalized as follows:

**Definition 3.** Two representations $\rho_1, \rho_2 : G \to GL(\mathbb{R}^c)$ are *equivalent* if there is a matrix $Q \in GL(\mathbb{R}^c)$ such that

$$\rho_2(g) = Q^{-1} \rho_1(g) Q \tag{31}$$

for all $g \in G$.

Intuitively, $\rho_1$ and $\rho_2$ differ only by a change of basis, which is given by $Q$.

### B.2 DECOMPOSITION INTO IRREDUCIBLE REPRESENTATIONS

We can now discuss irreducible representations, which play an important role for solving the kernel and PDO steerability constraints.

**Definition 4.** A linear subspace $W \subseteq \mathbb{R}^c$ is called *invariant* under a representation $\rho : G \to GL(\mathbb{R}^c)$ if $\rho(g)w \in W$ for all $g \in G$ and $w \in W$. In this case, we can define the restriction $\rho_{|W} : G \to GL(W)$, called a *subrepresentation* of $\rho$.

**Definition 5.** A representation $\rho$ is called *irreducible* if all its subrepresentations are either $\rho$ itself or representations $G \to GL(\{0\})$, where $\{0\}$ is the trivial vector space.

For example, $\rho_1$ and $\rho_2$ are both subrepresentations of $\rho_1 \oplus \rho_2$, so direct sums are never irreducible. A natural question is the converse: if a representation is *not* equivalent to a direct sum, does that mean that it is irreducible? In other words, can all representations be split into a direct sum of irreducible ones? In general, this is false, but it holds for the cases that interest us:

**Definition 6.** A *topological group* is a group $G$ equipped with a topology, such that the group multiplication $G \times G \to G$ and the inverse map $G \to G$ are continuous with respect to that topology. A *compact group* is a topological group that is compact as a topological space.

**Theorem 6.** *Let $G$ be a compact group. Then every finite-dimensional representation of $G$ is equivalent to a direct sum of irreducible representations.*

Since we only consider compact subgroups of $O(2)$ and $O(3)$, this theorem applies to all the cases we solve.

So we can always write

$$\begin{aligned} \rho_{\text{in}} &= Q_{\text{in}}^{-1} \bigoplus_{i \in I_{\text{in}}} \psi_i \; Q_{\text{in}} \\ \rho_{\text{out}} &= Q_{\text{out}}^{-1} \bigoplus_{i \in I_{\text{out}}} \psi_i \; Q_{\text{out}} \end{aligned} \tag{32}$$

where the $\psi_i$ are irreducible representations. It is then easy to show (Weiler & Cesa, 2019) that a kernel $k$ solves the $G$-steerability constraint

$$k(gx) = \rho_{\text{out}}(g)^{-1} k(x) \rho_{\text{in}}(g)^{-1} \tag{33}$$

if and only if $\kappa := Q_{\text{out}} k \, Q_{\text{in}}^{-1}$ solves a block-wise steerability constraint between irreducible representations. Concretely,

$$\kappa^{ij}(gx) = \psi_i(g)^{-1}\kappa^{ij}(x)\psi_j(g)^{-1} \qquad \forall i,j \tag{34}$$

where $\kappa^{ij}(x)$ is the submatrix of $\kappa(x)$ that belongs to $\psi_i$ and $\psi_j$.

The approach to solving the steerability constraint is thus to solve Eq. (34) for arbitrary irreducible representations $\psi_i$ and $\psi_j$. For general (not necessarily irreducible) $\rho_{\text{out}}$ and $\rho_{\text{in}}$, we then first find the decompositions into irreducible representations. Each basis element $\kappa^{ij}$ of the solution to Eq. (34) is then padded with zeros to the right size and finally transformed via $k = Q_{\text{out}}^{-1}\kappa Q_{\text{in}}$ to get the final basis elements. See Weiler & Cesa (2019) for a more detailed discussion and visualization.

Clearly, this procedure works just as well for PDOs as it does for kernels: for PDOs, we need to find the restriction of the kernel solution space to polynomials, and it does not matter whether we restrict on the level of irreducible representations and then combine them, or first combine irreducible representations and then restrict.

### B.3 Specific representations

We now define all the types of representations that occur in the main paper:

- As already mentioned in the paper, scalar fields are described by the *trivial representation* $\rho(g) := 1$ and vector fields by the representation $\rho(g) := g$ (for $G \leq \text{GL}(\mathbb{R}^d)$).

- For a finite group $G$, the *regular representation* is $\rho : G \to \text{GL}(\mathbb{R}^{|G|})$, defined by

$$\rho(g)e_{g'} := e_{gg'}, \tag{35}$$

  where $(e_g)_{g \in G}$ is the canonical basis of $\mathbb{R}^{|G|}$ (for some ordering of $G$). So this representation associates one basis vector to each group element and then acts by permuting these basis vectors. $\rho(g)$ is thus always a permutation matrix.

- *Quotient representations* are a generalization of regular representations, defined as follows: let $G$ be a finite group and $H \leq G$ a subgroup. Then we define the quotient representation $\rho_{\text{quot}}^{G/H} : G \to \text{GL}(\mathbb{R}^{|G|/|H|})$ by

$$\rho_{\text{quot}}^{G/H}(g)e_{g'H} := e_{gg'H}, \tag{36}$$

  where we now use a basis indexed by the cosets $gH \in G/H$ for $g \in G$. For $H = \{e\}$, we recover regular representations. Appendix C by Weiler & Cesa (2019) provides some intuition for these quotient representations in the case where $G$ and $H$ are both cyclic groups $C_N$ and $C_M$.

## C Intuition for the group action on polynomials

Our work makes heavy use of terms of the form $p(gx)$, where $p$ is a polynomial $p : \mathbb{R}^d \to \mathbb{R}$, $g \in G$ is a group element, and $x \in \mathbb{R}^d$. We would now like to provide a bit more intuition for this action of the group $G$ on polynomials. Note that we will only cover *scalar-valued* polynomials in this appendix, i.e. using trivial representations. The general case is straight-forward: the group acts on each polynomial in a matrix of polynomials the way we describe here, while $\rho_{\text{in}}(g)$ and $\rho_{\text{out}(g)}$ act via matrix multiplication.

To prevent confusion, let us reiterate that we can think of polynomials in two different ways. The first is as a formal expression, where $x$ is a placeholder for things to be plugged in. This is the approach we take when connecting polynomials to PDOs, where we plug in differential operators for $x$. The second is as a specific type of function on $\mathbb{R}^d$—in which case $x \in \mathbb{R}^d$ is simply the argument of that function.

This second perspective is the one in which the group action on polynomials is easiest to understand. Specifically, the action of $g$ on $x$ is simply matrix multiplication. Similar to how $x \mapsto p(x)$ defines a function on $\mathbb{R}^d$, $x \mapsto p(gx)$ defines a different function on $\mathbb{R}^d$. We could think of it as composing the function $p$ with the group action of $g$ on $\mathbb{R}^d$.

Crucially, this new function is still a polynomial in the components of $x$, just with different coefficients. For purposes of illustration, consider a very simple example with $d = 2$ and $G = \mathrm{SO}(2)$. We will use the polynomial $p(x) = x_2^2 + x_1$ (this is just meant to be a simple non-trivial polynomial, it does not have special equivariance properties).

We can represent $g$ as a rotation matrix parameterized by an angle $\theta$. The action on $x$ is then given by

$$gx = \begin{pmatrix} \cos\theta & -\sin\theta \\ \sin\theta & \cos\theta \end{pmatrix} \begin{pmatrix} x_1 \\ x_2 \end{pmatrix} = \begin{pmatrix} x_1\cos\theta - x_2\sin\theta \\ x_1\sin\theta + x_2\cos\theta \end{pmatrix} . \tag{37}$$

The left-hand side is what we now plug into our polynomial, which yields

$$p(gx) = (gx)_2^2 + (gx)_1 = (x_1\sin\theta + x_2\cos\theta)^2 + (x_1\cos\theta - x_2\sin\theta) . \tag{38}$$

We can expand this and collect the coefficients for each power of $x_1$ and $x_2$, which yields

$$p(gx) = \sin^2(\theta)x_1^2 + 2\sin(\theta)\cos(\theta)x_1x_2 + \cos^2(\theta)x_2^2 + \cos(\theta)x_1 - \sin(\theta)x_2 . \tag{39}$$

If desired, we can now switch to the first perspective, and interpret this polynomial a formal expression defined by its coefficients. In that perspective, $g$ acts on $p$ by modifying its coefficients, rather than by composition. Computing the new coefficients is straightforward analytically (given a matrix representation of $g$); we just gave a simple example for a polynomial of order two. The general case proceeds along the same lines, it just requires using the binomial theorem (or the multinomial theorem for $d > 2$).

## D  BACKGROUND ON DISTRIBUTIONS

In Appendix E, we will describe our framework for equivariant maps using convolutions with Schwartz distributions. To facilitate that, we now give the necessary background on Schwartz distributions. We restrict ourselves to what is absolutely necessary for our purposes; for a much more thorough introduction and for proofs, see e.g. Treves (1967).

### D.1  BASIC DEFINITIONS

**Definition 7.** For $U \subset \mathbb{R}^d$ open, $\mathcal{D}(U) := C_c^\infty(U)$ is called the space of *test functions* on $U$ ($C_c^\infty(U)$ is the space of compactly supported smooth functions $U \to \mathbb{R}$). A sequence $(\varphi_n)$ in $\mathcal{D}(U)$ is defined to converge to 0 iff

(i) there is a compact subset $K \subset U$ such that the support of each $\varphi_n$ is contained in $K$ and

(ii) $\partial^\alpha \varphi_n \to 0$ uniformly for all multi-indices $\alpha$.

One can define the so-called *canonical LF topology* on $\mathcal{D}(U)$. This topology induces the notion of convergence just given. However, constructing this topology explicitly is unnecessary for our purposes since knowing when sequences converge will be enough.

**Definition 8.** A linear functional $T : \mathcal{D}(U) \to \mathbb{R}$ is defined as continuous if for every sequence $\varphi_n$ that converges to 0 in $\mathcal{D}(U)$, $T\varphi_n \to 0$ (in $\mathbb{R}$).

Such a continuous functional is called a *distribution* on $U$. The space of all distributions on $U$ is written as $\mathcal{D}'(U)$.

This notion of continuity is also induced by the canonical LF topology and then $\mathcal{D}'(U)$ is the topological dual of $\mathcal{D}(U)$ as the notation suggests.

So intuitively, a distribution is a "reasonable" way of linearly assigning a number in $\mathbb{R}$ to each test function in $C_c^\infty$.

A function $f : U \to \mathbb{R}$ induces a distribution $T_f \in \mathcal{D}'(U)$, defined by

$$T_f\varphi := \int_U f\varphi \, d\lambda^d . \tag{40}$$

Since $\varphi$ has compact support, we don't need many restrictions on $f$ (for example, any locally integrable function $f \in L_{1,\text{loc}}(U)$ works).

We will also use the *duality pairing*

$$\langle T, \varphi \rangle := T(\varphi) \tag{41}$$

and under slight abuse of notation also write

$$\langle f, \varphi \rangle := T_f(\varphi) = \int_U f\varphi \, d\lambda^d \,. \tag{42}$$

Note that this coincides with the inner product on $L_2(U)$, but it allows functions $f$ that are not in $L_2$ while in exchange requiring $\varphi$ to have compact support.

## D.2 Convolutions

We define the translation $\tau_x : \mathbb{R}^d \to \mathbb{R}^d$ by $\tau_x(y) = y + x$ and set $\tau_x f := f \circ \tau_{-x}$ for functions $f$ on $\mathbb{R}^d$. This is exactly the same as the action of $(\mathbb{R}^d, +)$ we used in the main paper, just with more explicit notation. Furthermore, we write $\check{f}(x) := f(-x)$.

Then we can define the convolution between a distribution and a test function: for $f \in \mathcal{D}(U)$ and $T \in \mathcal{D}'(U)$, the convolution $T * f \in C^\infty(U)$ is defined by

$$(T * f)(x) := T(\tau_x \check{f}) \,. \tag{43}$$

Explicitly, $(\tau_x \check{f})(y) = f(x - y)$. This immediately shows that this notion of convolution extends the classical one, i.e.

$$T_g * f = g * f \,. \tag{44}$$

## D.3 Derivatives

**Definition 9.** For $T \in \mathcal{D}'(U)$, we define the distributional derivative $\partial^\alpha T$ as the unique distribution on $U$ for which

$$\langle \partial^\alpha T, \varphi \rangle = (-1)^{|\alpha|} \langle T, \partial^\alpha \varphi \rangle \,. \tag{45}$$

The distributional derivative of all orders always exists, i.e. "distributions are infinitely differentiable", but of course only in this distributional sense. At least for $f \in C^\infty(U)$ (but also under much weaker assumptions), this definition also extends the definition of derivatives of functions, in the sense that $T_{\partial^\alpha f} = \partial^\alpha T_f$.

## D.4 Composition with diffeomorphisms

Let $F : U \to U$ be a diffeomorphism and $T \in \mathcal{D}'(U)$. Then we define the composition $T \circ F \in \mathcal{D}'(U)$ as

$$\langle T \circ F, \varphi \rangle := \langle T, \left| \det DF^{-1} \right| \varphi \circ F^{-1} \rangle \tag{46}$$

where $\det DF^{-1}$ is the inverse Jacobian.

As in the previous constructions, this extends the definition for classical functions in the sense that

$$T_{f \circ F} = T_f \circ F \,. \tag{47}$$

This follows immediately from the transformation theorem for integrals:

$$\langle T_{f \circ F}, \varphi \rangle = \int_U (f \circ F) \cdot \varphi \, d\lambda \tag{48}$$

$$= \int_U f \cdot (\varphi \circ F^{-1}) \left| \det DF^{-1} \right| d\lambda \tag{49}$$

$$= \langle T_f, (\varphi \circ F^{-1}) \left| \det DF^{-1} \right| \rangle \tag{50}$$

$$= \langle T_f \circ F, \varphi \rangle \,. \tag{51}$$

Furthermore, this type of composition is associative:

$$\langle (T \circ F) \circ G, \varphi \rangle = \langle T \circ F, |\det DG^{-1}| \varphi \circ G^{-1} \rangle \tag{52}$$

$$= \langle T, |\det DF^{-1}| |\det DG^{-1} \circ F^{-1}| \varphi \circ G^{-1} \circ F^{-1} \rangle \tag{53}$$

$$= \langle T, |\det(DG^{-1} \circ F^{-1}) DF^{-1}| \varphi \circ (F \circ G)^{-1} \rangle \tag{54}$$

$$= \langle T, |\det D(F \circ G)^{-1}| \varphi \circ (F \circ G)^{-1} \rangle \tag{55}$$

$$= \langle T \circ (F \circ G), \varphi \rangle . \tag{56}$$

We are particularly interested in the case where $F$ is a linear transformation, i.e. $F \in \mathrm{GL}(\mathbb{R}^d)$. Then we have

$$\langle T \circ F, \varphi \rangle = |\det F^{-1}| \langle T, \varphi \circ F^{-1} \rangle \tag{57}$$

(note that in this case, we can pull out the determinant because it is just a constant). For translations, we get

$$\langle T \circ \tau_x, \varphi \rangle = \langle T, \varphi \circ \tau_{-x} \rangle = \langle T, \tau_x \varphi \rangle . \tag{58}$$

### D.5 THE DIRAC DELTA DISTRIBUTION

A very simple but important distribution is the following:

**Definition 10.** For any $x \in \mathbb{R}^d$, the Dirac delta distribution $\delta_x \in \mathcal{D}(\mathbb{R}^d)$ is defined by

$$\delta_x[\varphi] := \varphi(x) . \tag{59}$$

It is clear from the definition that the distributional derivatives of the delta distribution are given by

$$(\partial^\alpha \delta_x)[\varphi] = (-1)^{|\alpha|} \partial^\alpha \varphi(x) . \tag{60}$$

### D.6 CONVERGENCE OF DISTRIBUTIONS

**Definition 11.** We say that a sequence $T_n$ of distributions converges to $T$, written $T_n \to T$, if it converges pointwise, i.e.

$$\langle T_n, \varphi \rangle \to \langle T, \varphi \rangle \quad \text{for } n \to \infty \tag{61}$$

for all $\varphi \in \mathcal{D}(U)$.

Abusing notation a bit, we also write $f_n \to T$ for functions $f_n$ if $T_{f_n} \to T$.

One important type of function sequences are *Dirac sequences*, which converge to the Delta distribution:

**Lemma 7.** *Let $f_n$ be a sequence in $L^1(\mathbb{R}^d)$ such that*

*(i)* $f_n \geq 0$,

*(ii)* $\|f_n\|_1 = 1$, *and*

*(iii)* $\int_{\mathbb{R}^d \setminus B_\varepsilon(0)} f_n d\lambda^d \to 0$ *for all $\varepsilon > 0$.*

*Then $f_n \to \delta_0$ in the sense of distributions.*

We also note that convergence plays together nicely with derivatives and with convolutions:

**Lemma 8.** *If $T_n \to T$, then $\partial^\alpha T_n \to \partial^\alpha T$ for all $\alpha$.*

*Proof.*

$$\langle \partial^\alpha T_n, \varphi \rangle = (-1)^{|\alpha|} \langle T_n, \partial^\alpha \varphi \rangle$$
$$\to (-1)^{|\alpha|} \langle T, \partial^\alpha \varphi \rangle \tag{62}$$
$$= \langle \partial^\alpha T, \varphi \rangle .$$

$\square$

**Lemma 9.** *If $T_n \to T$, then $T_n * f \to T * f$ pointwise for all $f \in \mathcal{D}(U)$.*

*Proof.*

$$
\begin{aligned}
(T_n * f)(x) &= T_n(\tau_x \check{f}) \\
&\to T(\tau_x \check{f}) \\
&= (T * f)(x) \, .
\end{aligned}
\tag{63}
$$

$\square$

### D.7 TEMPERED DISTRIBUTIONS AND THE FOURIER TRANSFORM

The *Schwartz space* $\mathcal{S}(\mathbb{R}^d)$ is the space of functions for which all derivatives decay very quickly as $|x| \to \infty$. More precisely, a smooth function $f : \mathbb{R}^d \to \mathbb{R}$ is in $\mathcal{S}(\mathbb{R}^d)$ iff

$$
\sup_{x \in \mathbb{R}^d} \left\| x^\beta \partial^\alpha f(x) \right\| < \infty \, .
\tag{64}
$$

Intuitively, all derivatives of $f$ must decay more quickly than any polynomial. Examples are compactly supported functions or Gaussians.

Its dual space $\mathcal{S}'(\mathbb{R}^d)$ is called the space of *tempered distributions* and can be continuously embedded into $\mathcal{D}'(\mathbb{R}^d)$. Tempered distributions are still very general; in particular the delta distribution, distributions induced by functions, and derivatives of tempered distributions are all tempered.

The reason we're interested in tempered distributions is that there is a Fourier transform defined by

$$
\langle \mathcal{F}\{T\}, \varphi \rangle := \langle T, \mathcal{F}\{\varphi\} \rangle
\tag{65}
$$

for tempered distributions $T$. This is an automorphism on $\mathcal{S}'(\mathbb{R}^d)$.

We use the following convention for the Fourier transform on functions:

$$
\mathcal{F}\{f\}(\xi) := (2\pi)^{-d/2} \int_{\mathbb{R}^d} f(x) \exp(-ix \cdot \xi) \, dx \, ,
\tag{66}
$$

which means the inverse is given by

$$
\mathcal{F}^{-1}\{f\}(x) := (2\pi)^{-d/2} \int_{\mathbb{R}^d} f(x) \exp(ix \cdot \xi) \, d\xi \, .
\tag{67}
$$

We will later need the Fourier transform of derivatives of the Dirac delta distribution:

$$
\begin{aligned}
\langle \mathcal{F}\{\partial^\alpha \delta_0\}, \varphi \rangle &= \langle \partial^\alpha \delta_0, \mathcal{F}\{\varphi\} \rangle \\
&= (-1)^{|\alpha|} \partial^\alpha \mathcal{F}\{\varphi\}\Big|_0 \\
&= (-i)^{|\alpha|} \mathcal{F}\{x^\alpha \varphi\}\Big|_0 \\
&= (-i)^{|\alpha|} (2\pi)^{-d/2} \int x^\alpha \varphi(x) \exp(-ix \cdot 0) dx \\
&= (-i)^{|\alpha|} (2\pi)^{-d/2} \int x^\alpha \varphi(x) dx \, ,
\end{aligned}
\tag{68}
$$

so $\mathcal{F}\{\partial^\alpha \delta_0\} \propto x^\alpha$ (or more precisely, the distribution induced by the function $x \mapsto x^\alpha$).

### D.8 MATRICES OF DISTRIBUTIONS

Analogously to how we defined matrices of PDOs or of polynomials, we will need matrices of distributions. We will write $\mathcal{D}'(U, \mathbb{R}^{c_{\text{out}} \times c_{\text{in}}})$ for the space of $c_{\text{out}} \times c_{\text{in}}$ dimensional matrices with entries in $\mathcal{D}'(U)$. We get a pairing

$$
\langle \cdot, \cdot \rangle : \mathcal{D}'(U, \mathbb{R}^{c_{\text{out}} \times c_{\text{in}}}) \times \mathcal{D}(U, \mathbb{R}^{c_{\text{in}}}) \to \mathbb{R}^{c_{\text{out}}}
\tag{69}
$$

defined by

$$\langle T, f \rangle_i := \sum_j \langle T_{ij}, f_j \rangle \tag{70}$$

The convolution of $T \in \mathcal{D}'(U, \mathbb{R}^{c_{out} \times c_{in}})$ and $f \in \mathcal{D}(U, \mathbb{R}^{c_{in}})$ is defined analogously to the scalar case, i.e. $(T * f)(x) := \langle T, \tau_x \check{f} \rangle$, where we now use the more general duality pairing just defined.

We can also define the multiplication of matrices of distributions by matrices over $\mathbb{R}$. For a matrix $A \in \mathbb{R}^{c_{out} \times c_{out}}$ and a matrix of distributions $T \in \mathcal{D}'(\mathbb{R}^d, \mathbb{R}^{c_{out} \times c_{in}})$, we write

$$(AT)_{ij} := \sum_l A_{il} T_{lj} \in \mathcal{D}'(\mathbb{R}^d, \mathbb{R}^{c_{out} \times c_{out}}) \tag{71}$$

(multiplying a distribution by a scalar obviously defines another distribution). Analogously, we define $TB$ for $B \in \mathbb{R}^{c_{in} \times c_{in}}$. It immediately follows that

$$A\langle T, B\varphi \rangle = \langle ATB, \varphi \rangle \tag{72}$$

holds.

Composition with diffeomorphisms of $\mathbb{R}^d$ can simply be defined component-wise for matrices of distributions. This commutes with multiplication by constant matrices, meaning that

$$ATB \circ F = A(T \circ F)B. \tag{73}$$

Finally, we also define the Fourier transform component-wise, and this commutes with multiplication by matrices in the same way.

# E  DISTRIBUTIONAL FRAMEWORK FOR EQUIVARIANT MAPS

In this section, we develop two main results: first that all linear continuous translation equivariant maps between feature spaces are convolutions with some distribution, and then the equivariance constraint for such convolutions. See Appendix D for background on distributions. We equip $\mathcal{D}(U)$ with the canonical LF topology throughout this section.

## E.1  TRANSLATION EQUIVARIANT MAPS ARE CONVOLUTIONS WITH DISTRIBUTIONS

We begin by showing that the framework using convolutions with Schwartz distributions encompasses all translation equivariant continuous linear maps between feature spaces. As preparation, we prove a simple Lemma on the reflection map $s$:

**Lemma 10.** *The map $s : \mathcal{D}(U) \to \mathcal{D}(U)$ given by $s(f) := \check{f}$ is linear and continuous.*

*Proof.* Linearity is clear:

$$s(af + bg)(x) = af(-x) + bg(-x) = as(f)(x) + bs(g)(x). \tag{74}$$

For continuity, we use the fact that a linear map from $\mathcal{D}(U)$ to itself is continuous if and only if it is sequentially continuous (Treves, 1967, Proposition 14.7). So take any sequence $f_n \to 0$ in $\mathcal{D}(U)$. This means that

    (i) there is a compact subset $K \subset U$ such that the support of each $f_n$ is contained in $K$

    (ii) $\partial^\alpha f_n \to 0$ uniformly for all multi-indices $\alpha$

We set $-K := \{-x \,|\, x \in K\}$, then the support of $s(f_n)$ (which is just the mirror of the support of $f_n$) is contained in $-K$, and $-K$ is compact. Additionally,

$$\partial^\alpha s(f_n) = (-1)^\alpha s\left(\partial^\alpha f_n\right). \tag{75}$$

Since $\partial^\alpha f_n \to 0$ uniformly, the same is true for $s\left(\partial^\alpha f_n\right)$. It follows that $s$ is sequentially continuous, and thus continuous, which concludes the proof. $\square$

We first prove our main generality result for the special case of one-dimensional fibers:

**Lemma 11.** *Let $\Phi : \mathcal{D}(U) \to \mathcal{D}(U)$ be a translation equivariant continuous linear map. Then there is a distribution $T \in \mathcal{D}'(U)$ such that $\Phi(f) = T * f$.*

*Proof.* Let $\Phi : \mathcal{D}(U) \to \mathcal{D}(U)$ be any continuous linear map, where continuity is understood with respect to the canonical LF topology. We then define

$$T : \mathcal{D}(U) \to \mathbb{R}, \quad T(f) := \Phi(\check{f})(0). \tag{76}$$

Equivalently, we can write $T = \delta_0 \circ \Phi \circ s$, understood as normal composition of functions. $\delta_0$ and $\Phi$ are linear and continuous, as is $s$ by Lemma 10. Therefore, $T$ is also linear and continuous, and thus a Schwartz distribution $T \in \mathcal{D}'(U)$.

We will also need that

$$
\begin{aligned}
\left( \widetilde{\tau_{-x} f} \right)(y) &= (\tau_{-x} f)(-y) \\
&= f(x - y) \\
&= \check{f}(y - x) \\
&= (\tau_x \check{f})(y).
\end{aligned}
\tag{77}
$$

Now using the assumption that $\Phi$ is translation equivariant, i.e.

$$\tau_x \circ \Phi = \Phi \circ \tau_x, \tag{78}$$

it follows that convolution with $T$ is given by

$$
\begin{aligned}
(T * f)(x) &= \langle T, \tau_x \check{f} \rangle \\
&= \langle T, \widetilde{\tau_{-x} f} \rangle \\
&= \Phi(\tau_{-x} f)(0) \\
&= \tau_{-x} \Phi(f)(0) \\
&= \Phi(f)(x).
\end{aligned}
\tag{79}
$$

This shows that convolution with $T$ is $\Phi$, which concludes the proof. $\qquad\square$

Finally, we generalize to multi-dimensional feature fields:

**Theorem 12.** *Let $\Phi : \mathcal{D}(U, \mathbb{R}^{c_{in}}) \to \mathcal{D}(U, \mathbb{R}^{c_{out}})$ be a translation equivariant continuous linear map. Then there is a distribution $T \in \mathcal{D}'(U, \mathbb{R}^{c_{out} \times c_{in}})$ such that $\Phi(f) = T * f$.*

*Proof.* We write $e_j$ for the $j$-th canonical basis vector of $\mathbb{R}^{c_{in}}$. For $f \in \mathcal{D}(U, \mathbb{R}^{c_{in}})$, we write $f_j$ for its $j$-th component. Then we have

$$\Phi(f) = \Phi \left( \sum_{j=1}^{c_{in}} f_j e_j \right) = \sum_{j=1}^{c_{in}} \Phi\left(f_j e_j\right). \tag{80}$$

We now define the components $\Phi_{ij} : \mathcal{D}(U) \to \mathcal{D}(U)$ by

$$\Phi_{ij}(g) = \left(\Phi(g e_j)\right)_i. \tag{81}$$

Here, $g e_j \in \mathcal{D}(U, \mathbb{R}^{c_{in}})$ is the map with $g$ in its $j$-th output component and 0 in the other ones. We can then write

$$\Phi(f)_i = \sum_{j=1}^{c_{in}} \Phi(f_j e_j)_i = \sum_{j=1}^{c_{in}} \Phi_{ij}(f_j). \tag{82}$$

It is clear that the components $\Phi_{ij}$ of $\Phi$ are still translation equivariant, continuous and linear. Therefore, by the previous Lemma, there are distributions $T_{ij} \in \mathcal{D}'(U)$ such that $\Phi_{ij}(g) = T_{ij} * g$. We then get

$$\Phi(f)_i = \sum_{j=1}^{c_{in}} T_{ij} * f_j = T * f \tag{83}$$

where $T \in \mathcal{D}'(U, \mathbb{R}^{c_{out} \times c_{in}})$ is the matrix of distributions with entries $T_{ij}$. $\qquad\square$

### E.2 EQUIVARIANCE CONSTRAINT FOR DISTRIBUTIONS

We will now characterize the distributions $T \in \mathcal{D}'(\mathbb{R}^d, \mathbb{R}^{c_{out} \times c_{in}})$ for which the operator

$$T_* : \mathcal{D}(\mathbb{R}^d, \mathbb{R}^{c_{in}}) \rightarrow C^\infty(\mathbb{R}^d, \mathbb{R}^{c_{out}}) \tag{84}$$

given by $f \mapsto T * f$ is $H$-equivariant. The codomain of $T_*$ may contain functions that are not compactly supported for some $T$. We are mostly interested in distributions $T$ for which the codomain can be restricted to $\mathcal{D}(\mathbb{R}^d, \mathbb{R}^{c_{out}})$ but for the discussion of equivariance this does not make any difference, so we keep the derivation general by not restricting the distribution $T$.

First, we can note that convolution with a distribution is always translation equivariant:

**Proposition 13.** *The map $f \mapsto T * f$ is translation equivariant for any distribution $T$.*

*Proof.* The general equivariance condition is

$$(h \cdot (T * f))(x) = (T * (h \cdot f))(x) \tag{85}$$

for all $f \in \mathcal{D}(\mathbb{R}^d, \mathbb{R}^{c_{in}}), h \in H, x \in \mathbb{R}^d$. More explicitly, this means

$$\rho_{\text{out}}(g)\langle T, \tau_{h^{-1}x}\check{f}\rangle = \langle T, \tau_x(\rho_{\text{in}}(g)\widetilde{f \circ h^{-1}})\rangle, \tag{86}$$

where $g$ is the linear component of $h$.

Let $h = t \in \mathbb{R}^d$, i.e. a pure translation. The condition then becomes

$$\langle T, \tau_{(x-t)}\check{f}\rangle = \langle T, \tau_x(\widetilde{f \circ \tau_{-t}})\rangle = \langle T, \tau_x\tau_{-t}\check{f}\rangle, \tag{87}$$

which always holds (since $\tau_{(x-t)} = \tau_x\tau_{-t}$). So $T_*$ is translation equivariant by construction. $\square$

Therefore, it suffices to consider pure linear transformations $g \in \mathrm{GL}(\mathbb{R}^d)$. For those, we have the following result, which generalizes the steerability constraint for PDOs and the one for kernels:

**Theorem 14.** *Let $T \in \mathcal{D}'(\mathbb{R}^d, \mathbb{R}^{c_{out} \times c_{in}})$. Then the map $f \mapsto T * f$ is $G$-equivariant if and only if*

$$T \circ g = |\det g|^{-1} \rho_{out}(g)T\rho_{in}(g)^{-1}. \tag{88}$$

As is shown in Appendix D, the definitions of convolution with distributions and of composition with a diffeomorphism are compatible with those for classical functions. So if $T$ is a classical kernel, then the constraint on $T$ is given by the same equation, which already gives us the steerability constraint for kernels.

*Proof.* We start with the constraint Eq. (86) from above with $h = g \in G$ and will transform this into the desired form Eq. (88). First, to simplify notation, we can remove all the reflections $\check{\cdot}$ in Eq. (86). This is now possible because $\widetilde{f \circ g^{-1}} = \check{f} \circ g^{-1}$, which was not true for translations. Furthermore, we can pull the translations to the other side of the duality pairing:

$$\rho_{\text{out}}(g)\langle T \circ \tau_{g^{-1}x}, f\rangle \overset{!}{=} \langle T \circ \tau_x, \rho_{\text{in}}(g)f \circ g^{-1}\rangle \tag{89}$$
$$= |\det g|\langle T \circ \tau_x \circ g, \rho_{\text{in}}(g)f\rangle.$$

Now we'll use

$$\tau_x \circ g = g \circ \tau_{g^{-1}x}, \tag{90}$$

since

$$x + gy = g(g^{-1}x + y). \tag{91}$$

Plugging this into Eq. (89), we get

$$\rho_{\text{out}}(g)\langle T \circ \tau_{g^{-1}x}, f\rangle \overset{!}{=} |\det g|\langle T \circ g \circ \tau_{g^{-1}x}, \rho_{\text{in}}(g)f\rangle. \tag{92}$$

We want to have only $f$ on the right side of the duality pairing, so we use the notation we introduced for multiplying distributions by matrices and get

$$\langle \rho_{\text{out}}(g)T \circ \tau_{g^{-1}x}, f\rangle \overset{!}{=} |\det g|\langle T \circ g \circ \tau_{g^{-1}x}\rho_{\text{in}}(g), f\rangle. \tag{93}$$

This holds for all $f$ iff we have equality of distributions,

$$\rho_{\text{out}}(g)T \circ \tau_{g^{-1}x} \overset{!}{=} |\det g| T \circ g \circ \tau_{g^{-1}x} \rho_{\text{in}}(g). \tag{94}$$

Finally, multiplication with matrices and composition with diffeomorphisms commute, so we can cancel the $\tau_{g^{-1}x}$. This means our final constraint on $T$ is

$$\rho_{\text{out}}(g)T \overset{!}{=} |\det g|(T \circ g)\rho_{\text{in}}(g), \tag{95}$$

which we can slightly rewrite as

$$T \circ g \overset{!}{=} |\det g|^{-1}\rho_{\text{out}}(g)T\rho_{\text{in}}(g)^{-1}. \tag{96}$$

This is exactly Eq. (88). All steps of the derivation work equally well in the other direction, which proves the "if and only if". $\qquad\square$

### E.3 DIFFERENTIAL OPERATORS AS CONVOLUTIONS

It is clear that convolutions with classical kernels are a special case of convolutions with distributions (see Appendix D). But we have claimed that convolutions with distributions also cover PDOs, which is what we show now.

We have already seen in Appendix D that derivatives of the delta distribution are closely related to PDOs. So we calculate the convolution with such derivatives:

$$
\begin{aligned}
((\partial^\alpha \delta_0) * f)(x) &= \langle \partial^\alpha \delta_0, \tau_x \check{f} \rangle \\
&= (-1)^{|\alpha|} \partial^\alpha (\tau_x \check{f})(0) \\
&= (-1)^{|\alpha|} \tau_x (\partial^\alpha \check{f})(0) \\
&= \tau_x \left( \widetilde{\partial^\alpha f} \right)(0) \\
&= \left( \widetilde{\partial^\alpha f} \right)(-x) \\
&= \partial^\alpha f(x).
\end{aligned} \tag{97}
$$

Noting that the map $T \mapsto T_*$ is $\mathbb{R}$-linear, we see that $D(p)f = p(\partial)\delta_0 * f$ for polynomials $p$. It then also immediately follows that this is true for matrices of polynomials $P$ because

$$
\begin{aligned}
(D(P)f)_i &= \sum_{j=1}^{c_{\text{in}}} D(P_{ij})f_j \\
&= \sum_{j=1}^{c_{\text{in}}} P_{ij}(\partial)\delta_0 * f_j \\
&= P(\partial)\delta_0 * f.
\end{aligned} \tag{98}
$$

This shows that PDOs can be interpreted as convolutions with distributions, as claimed.

### E.4 THE FOURIER DUALITY BETWEEN KERNELS AND PDOS

Interpreting PDOs as convolutions with distributions allows us to relate them to classical convolutional kernels via a Fourier transform. We have already seen in Appendix D that

$$\mathcal{F}\{\partial^\alpha \delta_0\} = (2\pi)^{-d/2}(-i)^{|\alpha|}x^\alpha, \tag{99}$$

which by linearity of the Fourier transform immediately implies

$$\mathcal{F}\{P(\partial)\delta_0\}_{ij} = (2\pi)^{-d/2}\sum_\alpha (-i)^{|\alpha|}c_\alpha^{ij}x^\alpha. \tag{100}$$

for $P_{ij} = \sum_\alpha c_\alpha^{ij}x^\alpha$. So the Fourier transform of derivatives of the delta distribution are polynomials, and of course vice versa via the inverse Fourier transform. Since PDOs are convolutions with such delta distribution derivatives, we can also interpret them as *convolutions with the (inverse) Fourier transform of polynomials*. We will use this interpretation to give a derivation of the PDO steerability constraint that sheds some light on the similarities and differences to the kernel steerability constraint.

We begin by proving a basic fact about the Fourier transform of a composition of functions:

**Lemma 15.** *Let $g \in \mathrm{GL}(\mathbb{R}^d)$ and $\varphi \in \mathcal{D}(U)$. Then*

$$\mathcal{F}\{\varphi \circ g\} = \left|\det g^{-1}\right| \mathcal{F}\{\varphi\} \circ g^{-T}, \tag{101}$$

*where we use the shorthand $g^{-T} := \left(g^{-1}\right)^T$.*

*Proof.* First we apply the transformation theorem for integrals:

$$\mathcal{F}\{\varphi \circ g\}(\xi) = (2\pi)^{-d/2} \int \varphi(gx) \exp(-ix \cdot \xi) dx$$
$$= \left|\det g^{-1}\right| (2\pi)^{-d/2} \int \varphi(x) \exp(-ig^{-1}x \cdot \xi) dx. \tag{102}$$

Now we just rewrite $g^{-1}x \cdot \xi = x \cdot g^{-T}\xi$, which gives the desired result. $\qquad\square$

This result holds more generally for tempered distributions (a subset of distributions for which the Fourier transform can be defined, see Appendix D):

**Proposition 16.** *For a tempered distribution $T$ and $g \in \mathrm{GL}(\mathbb{R}^d)$,*

$$\mathcal{F}\{T \circ g\} = \left|\det g^{-1}\right| \mathcal{F}\{T\} \circ g^{-T}. \tag{103}$$

*Proof.*

$$\begin{aligned}
\langle \mathcal{F}\{T \circ g\}, \varphi \rangle &= \langle T \circ g, \mathcal{F}\{\varphi\} \rangle \\
&= \left|\det g^{-1}\right| \langle T, \mathcal{F}\{\varphi\} \circ g^{-1} \rangle \\
&\overset{(1)}{=} \langle T, \mathcal{F}\{\varphi \circ g^T\} \rangle \\
&= \langle \mathcal{F}\{T\}, \varphi \circ g^T \rangle \\
&= \left|\det g^{-1}\right| \langle \mathcal{F}\{T\} \circ g^{-T}, \varphi \rangle,
\end{aligned} \tag{104}$$

where we used Lemma 15 for (1). $\qquad\square$

Now note that the equivariance constraint for distributions, Eq. (88), is equivalent to the constraint we get when we take the Fourier transform on both sides. That's because the Fourier transform is an automorphism on the space of tempered distributions. By applying the result we just proved, we then get the constraint

$$\left|\det g^{-1}\right| \mathcal{F}\{T\} \circ g^{-T} = \left|\det g^{-1}\right| \rho_{\mathrm{out}}(g) \mathcal{F}\{T\} \rho_{\mathrm{in}}(g)^{-1}. \tag{105}$$

We can cancel the determinants, which gives the equivariance constraint in Fourier space:

$$\mathcal{F}\{T\} \circ g^{-T} = \rho_{\mathrm{out}}(g) \mathcal{F}\{T\} \rho_{\mathrm{in}}(g)^{-1}. \tag{106}$$

Now let $T = P(\partial)\delta_0$, so that $T * f = D(P)f$ for some matrix of polynomials $P$. We've seen above that in this case $\mathcal{F}\{T\} = P \circ m_{-i}$ (up to a constant coefficient), where $m_{-i}$ is multiplication by the negative imaginary unit $-i$. So $D(P)$ is equivariant iff

$$P \circ m_{-i} \circ g^{-T} = \rho_{\mathrm{out}}(g) P \circ m_{-i} \rho_{\mathrm{in}}(g)^{-1}, \tag{107}$$

but since $m_{-i}$ is invertible and commutes with the other maps, we can cancel it and get the equivariance condition

$$P \circ g^{-T} = \rho_{\mathrm{out}}(g) P \rho_{\mathrm{in}}(g)^{-1}, \tag{108}$$

which is precisely the PDO steerability constraint. The reason that it differs slightly from the kernel steerability constraint can now be traced back to Lemma 15. Intuitively speaking, since PDOs are in a sense the Fourier transform of convolutional kernels, they transform differently under $\mathrm{GL}(\mathbb{R}^d)$, which leads to superficial differences in the steerability constraints. However, *Fourier transforms commute with rotations and reflections* (i.e. transformations from $\mathrm{O}(d)$), which is why for $G \leq \mathrm{O}(d)$, the two steerability constraints coincide.

### E.5 PDOs as the infinitesimal limit of kernels

Let $\psi_n$ be any sequence of functions such that $\psi_n \to \delta_0$, for example a Dirac sequence. Then for a polynomial $p = \sum_\alpha c_\alpha x^\alpha$, we also have

$$p(\partial)\psi_n \to p(\partial)\delta_0 \tag{109}$$

because of Lemma 8. Then Lemma 9 implies that

$$p(\partial)\psi_n * f \to p(\partial)\delta_0 * f = D(p)f \,, \tag{110}$$

where the convergence is understood pointwise. Therefore, any sequence of kernels that approximates the delta distribution (by becoming "increasingly narrow") can be used to approximate arbitrary PDOs by convolving with the derivatives of the kernels.

One example is a sequence of Gaussians

$$\psi_\varepsilon(x) := \frac{1}{(2\pi\varepsilon)^{d/2}} e^{-x^2/\varepsilon} \tag{111}$$

(now indexed by $\varepsilon > 0$ instead of natural numbers). For $\varepsilon \to 0$, we have $\psi_\varepsilon \to \delta_0$, as is easy to check with Lemma 7. This naturally leads to the "derivative of Gaussian" discretization used in our experiments: we can approximate the PDO as convolution with a derivative of a Gaussian kernel and then simply discretize this Gaussian derivative by sampling it on the grid points.

The discussion in this subsection focused on $1 \times 1$ PDOs and kernels, i.e. $c_{\text{in}} = c_{\text{out}} = 1$, but since convergence for multi-dimensional PDOs and kernels works component-wise, everything generalizes immediately to that setting.

## F  The relation between kernels and PDOs

Convolutional kernels and PDOs are closely related but also differ in some important ways. This appendix is meant to briefly summarize various aspects of their relation that would otherwise only be scattered throughout this paper.

First and foremost, we would like to emphasize that, in a continuous setting, PDOs are *not* just a special case of convolutions with classical kernels. For example, the gradient operator is not represented by convolution with any kernel. In fact, even the identity operator (a zeroth-order PDO) is not the convolution with any function.

However, there are two caveats to the above statement. First, if we allow convolutions with *Schwartz distributions*, rather than the usual kernels, then this generalizes both PDOs and convolutions with functions. In this broader framework, we *can* thus interpret PDOs as convolutions. But to the best of our knowledge, convolutions with Schwartz distributions have not been previously discussed in a deep learning context, so for the common usage of "convolution", PDOs are distinct operators.

The second caveat is that when we discretize (translation-equivariant) PDOs on a regular grid, they become convolutions in this discrete setting. Even so, the differences in the continuum can matter in practice. First, we might want to discretize on point clouds or meshes. Second, even on a regular grid, there are different methods for discretizing PDOs. While all of them lead to convolutions, they lead to convolutions with slightly different kernels. This shows that there is no clear one-to-one correspondence between kernels and PDOs even on discrete regular grids. Finally, the space of discretizations of *equivariant* PDOs is not necessarily the same as the space of discretizations of *equivariant* kernels.

Finally, we would like to mention a connection that is outside the scope of this paper: *infinite series* of differential operators and convolutions can be the same even in the continuous setting. As an example, consider the diffusion equation

$$\partial_t u(x,t) = \Delta u(x,t) \,. \tag{112}$$

Its solution can be written as

$$u(x,t) = \exp(t\Delta)u(x,t=0) \,. \tag{113}$$

The time evolution operator $\exp(t\Delta)$ is an infinite series of differential operators, but can also be written as a convolution with a heat kernel. However, these infinite series are not covered by our work—we restrict ourselves to PDOs of *finite* order. We mention this connection between infinite PDOs and convolutions only to avoid confusion for readers familiar with this correspondence.

## G    PROOF OF THE PDO EQUIVARIANCE CONSTRAINT

In this section, we prove Proposition 1 and Theorem 2, the characterizations of translation- and $G$-equivariance for PDOs. Appendix E already contains a proof of Theorem 2 that is arguably more insightful, but in this section we present an elementary proof that does not rely on Schwartz distributions.

### G.1    TRANSLATION EQUIVARIANCE

We begin by proving that translation equivariance of a PDO is equivalent to spatially constant coefficients, Proposition 1.

Let $\Phi_{ij} = \sum_\alpha c_\alpha^{ij} \partial^\alpha$, where $c_\alpha^{ij} \in C^\infty(\mathbb{R}^d)$ and $i, j$ index the $c_{\text{out}} \times c_{\text{in}}$ matrix describing the PDO. The equivariance condition Eq. (2) for the special case of translations is

$$\Phi(t \rhd_{\text{in}} f) = t \rhd_{\text{out}} \Phi(f) \quad \forall t \in \mathbb{R}^d, f \in \mathcal{F}_{\text{in}}. \tag{114}$$

Using the definition of the $\rhd$ action in Eq. (1), this becomes

$$\sum_\alpha c_\alpha(x) \partial^\alpha f(x-t) = \sum_\alpha c_\alpha(x-t) \partial^\alpha f(x-t) \quad \forall x, t \in \mathbb{R}^d, f \in \mathcal{F}_{\text{in}}. \tag{115}$$

Here, $c_\alpha$ are matrix-valued, $f$ is vector valued, and we have a matrix-vector product between the two. Explicitly, this means

$$\sum_j \sum_\alpha c_\alpha^{ij}(x) \partial^\alpha f_j(x-t) = \sum_j \sum_\alpha c_\alpha^{ij}(x-t) \partial^\alpha f_j(x-t) \tag{116}$$

for all indices $i$. If $f$ is zero in all but one component, the sum over $j$ reduces to one summand. We therefore get a simpler scalar constraint

$$\sum_\alpha c_\alpha^{ij}(x) \partial^\alpha g(x-t) = \sum_\alpha c_\alpha^{ij}(x-t) \partial^\alpha g(x-t) \tag{117}$$

that has to hold for all indices $i, j$, where $g \in C^\infty(\mathbb{R}^d)$ is now scalar-valued.

Since this must hold for all functions $g$, we get an equality of differential operators,

$$\sum_\alpha c_\alpha^{ij}(x) \partial^\alpha = \sum_\alpha c_\alpha^{ij}(x-t) \partial^\alpha \qquad \forall x, t \in \mathbb{R}^d. \tag{118}$$

This implies $c_\alpha^{ij}(x-t) = c_\alpha^{ij}(x)$ for all $x, t$ and $\alpha$, which in turn means that $c_\alpha^{ij}$ must be constants. From Eq. (115), it is also apparent that constant coefficients are sufficient for translation equivariance, which proves the converse direction.

### G.2    $G$-EQUIVARIANCE

Secondly we prove Theorem 2, the $G$-steerability constraint for PDOs. Recall that the $G$-equivariance condition for $D(P)$ is

$$D(P)(g \rhd_{\text{in}} f) = g \rhd_{\text{out}} (D(P)f), \tag{119}$$

where $f \in \mathcal{F}_{\text{in}}$ and $g \in G$. If we write out the definition of the induced action $\rhd$, this becomes

$$D(P)(\rho_{\text{in}}(g)f \circ g^{-1}) = \rho_{\text{out}}(g)(D_P f) \circ g^{-1}. \tag{120}$$

We write $P = \sum_\alpha c_\alpha x^\alpha$, where $c_\alpha$ are constant matrix-valued coefficients $c_\alpha \in \mathbb{R}^{c_{\text{out}} \times c_{\text{in}}}$. The LHS of Eq. (120) is then

$$D(P)(\rho_{\text{in}}(g)f \circ g^{-1}) = \sum_\alpha c_\alpha \rho_{\text{in}}(g) \partial^\alpha (f \circ g^{-1}). \tag{121}$$

In Lemma 17, we will show that the last term in Eq. (121) is given by

$$\partial^\alpha (f \circ g^{-1}) = \left( \left( g^{-T} \partial \right)^\alpha f \right) \circ g^{-1}, \tag{122}$$

where $g^{-T} := \left(g^{-1}\right)^T$ and $\partial$ is understood as a vector that $g^{-T}$ acts on. Plugging this into Eq. (121), we get

$$D(P)(\rho_{\text{in}}(g)f \circ g^{-1}) = \sum_\alpha c_\alpha \left( \left(g^{-T}\partial\right)^\alpha \rho_{\text{in}}(g)f \right) \circ g^{-1} \tag{123}$$

for the LHS of Eq. (120). For the matrix of polynomials $P = \sum_\alpha c_\alpha x^\alpha$, we now define

$$g \cdot P := \sum_\alpha c_\alpha (g^{-1}x)^\alpha, \tag{124}$$

which is again a matrix of polynomials, each one rotated by $g$. Then Eq. (123) can be written more compactly as

$$D(P)(\rho_{\text{in}}(g)f \circ g^{-1}) = (D(g^T \cdot P)\rho_{\text{in}}(g)f) \circ g^{-1}. \tag{125}$$

Plugging this back into Eq. (120), canceling the $g^{-1}$ and using the fact that this has to hold for all $f$, we get

$$D(g^T \cdot P)\rho_{\text{in}}(g) = \rho_{\text{out}}(g)D(P) \tag{126}$$

as an equality of differential operators. We move the $\rho_{\text{in}}(g)$ to the other side and use the fact that the map $D$ is bijective, which yields our final constraint

$$P(g^{-T}x) = \rho_{\text{out}}(g)P(x)\rho_{\text{in}}(g)^{-1}. \tag{127}$$

This concludes the proof of the $G$-steerability constraint for PDOs.

Finally, we prove the Lemma we just made use of, a higher-dimensional chain rule for the special case we need:

**Lemma 17.** *Let $f \in C^\infty(\mathbb{R}^d, \mathbb{R}^c)$ and $g \in \text{GL}(\mathbb{R}^d)$. Then for any multi-index $\alpha$,*

$$\partial^\alpha(f \circ g^{-1}) = \left( \left(g^{-T}\partial\right)^\alpha f \right) \circ g^{-1}, \tag{128}$$

*where $g^{-T} := \left(g^{-1}\right)^T$.*

*Proof.* In general, for a linear map $A \in \text{GL}(\mathbb{R}^d)$, we have

$$\partial_i A_j(x) = A_{ji} \quad \forall x. \tag{129}$$

Therefore,

$$\partial_i(f \circ g^{-1}) = \sum_j (\partial_j f) \circ g^{-1} \partial_i(g^{-1})_j \tag{130}$$

$$= \sum_j (\partial_j f) \circ g^{-1} \cdot (g^{-1})_{ji} \tag{131}$$

$$= \left( \left( \left(g^{-1}\right)^T \partial \right)_i f \right) \circ g^{-1}. \tag{132}$$

We can apply this iteratively to show that

$$\partial^\alpha(f \circ g^{-1}) = \left( \left( \left(g^{-1}\right)^T \partial \right)^\alpha f \right) \circ g^{-1}. \tag{133}$$

$\square$

## H  TRANSFERRING STEERABLE KERNEL BASES TO STEERABLE PDO BASES

In this section, we develop the method presented in Section 3.2 in more detail and with proofs.

We fix a group $G \leq \text{O}(d)$ and representations $\rho_{\text{in}}$ and $\rho_{\text{out}}$. Then we write $\mathcal{K}$ for the space of $G$-steerable kernels. Because the steerability constraints for kernels and PDOs are identical in this setting, the space of equivariant PDOs is the image under the isomorphism $D$ of the intersection

$$\mathcal{K}_{\text{pol}} := \mathbb{R}[x_1, \ldots, x_d] \cap \mathcal{K}. \tag{134}$$

In words, the space of equivariant PDOs is isomorphic to the space of *polynomial* steerable kernels. Both spaces are infinite-dimensional real vector spaces. The question we tackle now is how we can find a basis of $\mathcal{K}_{\text{pol}}$ given a basis of $\mathcal{K}$ under certain conditions.

It will vastly simplify our discussion to treat $\mathcal{K}$ and $\mathcal{K}_{\text{pol}}$ as *modules* over invariant kernels instead of as real vector spaces, at least for now. A module is a generalization of a vector space, where the scalars for scalar multiplication can form a ring instead of a field. Because the radial parts of steerable kernels are unrestricted, it makes sense to think of $\mathcal{K}$ as a module over the ring of radial functions. Formally:

**Lemma 18.** $\mathcal{K}$ *is a* $C^\infty(\mathbb{R}_{\geq 0})$*-module, with scalar multiplication defined by*

$$(f\kappa)(x) := f\left(|x|^2\right)\kappa(x) \tag{135}$$

*for* $\kappa \in \mathcal{K}$ *and* $f \in C^\infty(\mathbb{R}_{\geq 0})$*. Similarly,* $\mathcal{K}_{pol}$ *is an* $\mathbb{R}[r^2]$*-module, where* $r^2 := x_1^2 + \ldots + x_d^2$ *and multiplication is simply multiplication of polynomials.*

*Proof.* The steerability constraint

$$\kappa(gx) = \rho_{\text{out}}(g)\kappa(x)\rho_{\text{in}}(g)^{-1} \tag{136}$$

is clearly $\mathbb{R}$-linear and in particular, $\mathcal{K}$ is closed under addition and $0 \in \mathcal{K}$. Furthermore, for $\kappa \in \mathcal{K}$,

$$\begin{aligned}
(f\kappa)(gx) &= f(|gx|^2)\kappa(gx) \\
&= f(|x|^2)\rho_{\text{out}}(g)\kappa(x)\rho_{\text{in}}(g)^{-1} \\
&= \rho_{\text{out}}(g)(f\kappa)(x)\rho_{\text{in}}(g)^{-1},
\end{aligned} \tag{137}$$

so $\mathcal{K}$ is also closed under the given scalar multiplication. The proof for $\mathcal{K}_{\text{pol}}$ is exactly analogous. $\square$

In Lemma 18 and in the following, we write $r^2$ instead of $|x|^2$ simply to emphasize its role as a polynomial; so when reading $r^2$, think of it as a polynomial, and when reading $|x|^2$ simply as a function of $x$.

A basis of a module is defined analogously to a basis of a vector space, as a set of linearly independent vectors that span the entire module. However, linear combinations now allow coefficients in the ring of radial functions, instead of only real numbers. This means that fewer vectors are needed to span the entire space, because the coefficients "do more work".

In contrast to vector spaces, not every module has a basis. $\mathcal{K}$ and $\mathcal{K}_{\text{pol}}$ do have a basis in the cases we consider in the paper but for this section that doesn't matter to us: we will simply assume that $\mathcal{K}$ has a basis with certain properties and then transfer this basis to $\mathcal{K}_{\text{pol}}$. That the method developed here is indeed applicable to subgroups of $O(2)$ and $O(3)$ will be the topic of Appendix J.

We roughly proceed in two steps:

1. We show that a basis of $\mathcal{K}$ (over $C^\infty(\mathbb{R}_{\geq 0})$) that consists only of polynomials (and fulfills a few other technical conditions) is also a basis of $\mathcal{K}_{\text{pol}}$, but this time of course over $\mathbb{R}[r^2]$.

2. We then show how to turn this *module basis* into a *vector space* basis of $\mathcal{K}_{\text{pol}}$.

The first step is formalized as follows:

**Proposition 19.** *Let* $B \subset \mathbb{R}[x_1, \ldots, x_d]^{c_{out} \times c_{in}}$ *be a basis of* $\mathcal{K}$ *such that no matrix of polynomials in* $B$ *is divisible by* $r^2$ *and each one is homogeneous. Then* $B$ *is also a basis of* $\mathcal{K}_{pol}$ *as an* $\mathbb{R}[r^2]$*-module.*

Here, we say that $b \in \mathbb{R}[x_1, \ldots, x_d]^{c_{out} \times c_{in}}$ is divisible by $r^2$ if *every* component is divisible by $r^2$ (as a polynomial). We call it homogeneous if all its entries are homogeneous polynomials of the same degree.

*Proof.* Linear independence is obvious: $\mathbb{R}[r^2] \subset C^\infty(\mathbb{R}_{\geq 0})$, so if $B$ is linearly independent over $C^\infty(\mathbb{R}_{\geq 0})$, then it is also linearly independent over $\mathbb{R}[r^2]$.

To show that $B$ generates $\mathcal{K}_{\text{pol}}$, first let $p \in \mathcal{K}_{\text{pol}}$ be homogeneous of degree $l$. Because $p \in \mathcal{K}$, there are $f_i \in C^\infty(\mathbb{R}_{\geq 0})$ and $\kappa_i \in B$ such that

$$p(x) = \sum_i f_i(|x|^2)\kappa_i(x) \,. \tag{138}$$

We want to show that $f_i \in \mathbb{R}[z]$, i.e. that $f_i$ is a polynomial. Since each $\kappa_i$ is homogeneous of degree $l_i$, we get

$$\sum_i f_i(\lambda^2|x|^2)\lambda^{l_i}\kappa_i(x) = \lambda^l \sum_i f_i(|x|^2)\kappa_i(x) \tag{139}$$

because of $p(\lambda x) = \lambda^l p(x)$ for any $\lambda \geq 0$. Because the $\kappa_i$ are linearly independent, we must have

$$f_i(\lambda^2|x|^2) = \lambda^{l-l_i}f_i(|x|^2) \,. \tag{140}$$

Thus, $f_i$ is homogeneous of degree $\frac{l-l_i}{2}$ and as we show in Lemma 20 below, this implies

$$f_i(z) = cz^{(l-l_i)/2} \,, \tag{141}$$

or alternatively

$$f_i(|x|^2) = c_i|x|^{l-l_i} \,. \tag{142}$$

What remains to show is that $\frac{l-l_i}{2}$ is a natural number, i.e. that $l - l_i$ is even and non-negative. To prove this, we divide all the $i$ into two groups: those for which $l - l_i$ is even and those for which it is odd. Then we get an expression of the form

$$p(x) = |x|\sum_k a_k r^{2m_k}\kappa_k + \sum_k b_k r^{2n_k}\kappa_k \,. \tag{143}$$

Each summand is a rational function (the quotient of two polynomials), so both sums are rational functions. $p(x)$ is also rational, so the first term on the RHS has to be rational. This is only possible if $a_k = 0$ for all $k$, otherwise we could divide by the (rational) sum and should get a rational function, but $|x|$ is not rational. This shows that $l - l_i$ is even for all $i$.

Now let $l_{\max} := \max_i l_i$. Then

$$p(x) = \frac{\sum_i c_i|x|^{l_{\max}-l_i}\kappa_i(x)}{|x|^{l_{\max}-l}} \,. \tag{144}$$

It is not possible to cancel any terms: because $\kappa_i$ is not divisible by $r^2$ and for one $i$, $l_{\max} - l_i = 0$, the enumerator is not divisible by $r^2$ (as a polynomial). Since the denominator is a power of $r^2$, the fraction can't be simplified. But we know that $p(x)$ is a polynomial. Therefore, $l_{\max} \leq l$.

In summary, we've shown that

$$f_i(z) = c_i z^{(l-l_i)/2} \tag{145}$$

where $l - l_i$ is even and non-negative. Therefore, all $f_i$ are polynomials.

Now recall that we assumed $p$ to be homogeneous. But this is no significant restriction: we can write any polynomial as a sum of homogeneous polynomials, each of which can be written as a linear combination of the $\kappa_i$, as we just showed. Adding those up leads to a linear combination of the $\kappa_i$ for arbitrary polynomials. This complete the proof. □

**Lemma 20.** *Let $f : \mathbb{R}_{\geq 0} \to \mathbb{R}$ be homogeneous of degree $l \in \mathbb{R}$, meaning that $f(\lambda x) = \lambda^l x$ for all $\lambda \geq 0$. Then $f(z) = cz^l$ for some $c \in \mathbb{R}$ (and if $l \in \mathbb{N}$, then $f$ is a polynomial).*

Note that this Lemma does not hold in higher dimensions – in general there are many more homogeneous functions than polynomials!

*Proof.* For any $z \geq 0$,

$$f(z) = z^l f(1) \,, \tag{146}$$

which proves the claim by setting $c := f(1)$. □

In Appendix J, we will show that the construction of angular basis elements described in Section 3.2 leads to a basis $B$ of $\mathcal{K}$ with the properties required by Proposition 19. It then follows that this also defines a basis of $\mathcal{K}_{\text{pol}}$ as a module. We now come to the second step, turning this module basis into a vector space basis.

**Proposition 21.** *Let $B$ be a basis of $\mathcal{K}_{pol}$ as an $\mathbb{R}[r^2]$-module. Then*

$$\left\{ r^{2k}b \,\big|\, k \in \mathbb{N}_{\geq 0}, b \in B \right\} \tag{147}$$

*is a basis of $\mathcal{K}_{pol}$ as a real vector space.*

*Proof.* Let $v \in \mathcal{K}_{\text{pol}}$. By assumption, there are then basis vectors $b_1, \ldots, b_n \in B$ and coefficients $p_1, \ldots, p_n \in \mathbb{R}[r^2]$ such that

$$v = \sum_{i=1}^{n} p_i b_i \,. \tag{148}$$

For each $i = 1, \ldots, n$ there are also real coefficients $a_0^{(i)}, \ldots, a_{m_i}^{(i)}$ such that

$$p_i = \sum_{k=0}^{m_i} a_k^{(i)} r^{2k} \,. \tag{149}$$

Combining these equations, we get

$$v = \sum_{i=1}^{n} \sum_{k=0}^{m_i} a_k^{(i)} r^{2k} b_i \,. \tag{150}$$

This is a real linear combination of elements of the form $r^{2k}b$ for $b \in B$. So since $v$ was arbitrary, the set of such elements does indeed span $\mathcal{K}_{\text{pol}}$.

To prove linear independence, let

$$\sum_{i=0}^{n} a_i r^{2k_i} b_i = 0 \tag{151}$$

for some choice of real coefficients $a_i$. Now we only need to note that $a_i r^{2k_i} \in \mathbb{R}[r^2]$. So we can interpret this expression as a linear combination over $\mathbb{R}[r^2]$. Because $B$ is a basis, it follows that $a_i r^{2k_i} = 0$ for all $i$, and thus $a_i = 0$. That proves linear independence. $\qquad\square$

As a final note, we show that the condition in Proposition 19 that the basis elements are not divisible by $r^2$ is purely technical; any basis can easily be transformed into one that fulfills it in a canonical way, as formalized by the following lemma. We do not formally need this result anywhere but it may prove useful if this approach is extended to other groups $G$ because it clarifies which parts of the conditions in Proposition 19 are actually important.

**Lemma 22.** *Let $B \subset \mathbb{R}[x_1, \ldots, x_d]^{c_{out} \times c_{in}}$ be a basis of $\mathcal{K}$. For $b \in B$, we write $b = r^{2k}b'$, where $k$ is chosen maximally such that there is an (automatically unique) polynomial matrix $b' \in \mathbb{R}[x_1, \ldots, x_d]^{c_{out} \times c_{in}}$. Then $B' := \{b' \,|\, b \in B\}$ is a basis of $\mathcal{K}$ and no $b'$ is divisible by $r^2$. Furthermore, if $b$ is homogeneous, then so is $b'$.*

*Proof.* First, note that $b'$ is in fact well-defined: For $k = 0$, $b' = b$ works, while for $k > \frac{\deg(b)}{2}$, no fitting $b'$ exists[7]. So there is some maximal $k \geq 0$ with the desired property. $b'$ is clearly unique, namely $b' = |x|^{-2k}b$ as functions on $\mathbb{R}^d$. Furthermore, $b'$ is not divisible by $r^2$ because $k$ was chosen maximally.

$B'$ is also clearly a generating set: $\kappa = \sum_i f_i b_i = \sum_i (f_i r^{2k_i}) b_i'$.

To prove linear independence, assume that

$$\sum_i f_i b_i' = 0 \,. \tag{152}$$

---

[7]Here, the degree of a matrix of polynomials is the maximum of the degrees of all components (though in this case, even if $k$ is larger than the minimum degree, no $b'$ exists).

We can write $k := \max_i k_i$ and then get

$$0 = r^{2k} \sum_i f_i b_i' = \sum_i f_i r^{2(k-k_i)} b_i \,. \tag{153}$$

This means that

$$f_i r^{2(k-k_i)} = 0 \quad \forall i, \tag{154}$$

which implies $f_i = 0$ for all $i$, since $r^{2(k-k_i)}$ is non-zero everywhere except in the origin and $f_i$ is continuous. Therefore, $B'$ is linearly independent and hence a basis of $\mathcal{K}$. □

Now we can formulate Proposition 19 more generally: for any basis $B$ of $\mathcal{K}$ consisting only of homogeneous matrices of polynomials, the corresponding $B'$ is a basis of $\mathcal{K}_{\text{pol}}$.

## I    SOLUTIONS FOR IMPORTANT GROUPS

This and the next two appendices describe the solutions of the PDO equivariance constraint for subgroups of O(2) and O(3). In this appendix, we describe the general form of these solutions; we recommend readers who are not interested in all the details focus on this one. Appendix J contains proofs for some claims we make in this appendix where these proofs do not provide as much insight. Finally, Appendix K contains tables with concrete solutions, it is mainly relevant to the implementation of steerable PDOs.

### I.1    SOLUTIONS FOR SUBGROUPS OF O(2)

To solve the steerability constraint for all (compact) subgroups $G \leq O(2)$ and for arbitrary representations $\rho_{\text{in}}$ and $\rho_{\text{out}}$, Weiler & Cesa (2019) derive explicit bases only for irreducible representations, since the general case can then easily be computed (see Appendix B for details). This works exactly the same for PDOs as well.

The angular basis elements $\chi_\beta$ that they describe for irreps are all matrices with entries of the form $\cos(k\varphi)$ and $\sin(k\varphi)$, where $k$ differs between basis elements but is the same for all entries of one matrix. For example, for $G = SO(2)$, $\rho_{\text{in}}$ the frequency $n$ irrep, i.e. $\rho_{\text{in}}(g) = g^n$, and $\rho_{\text{out}}$ trivial, the angular basis elements are

$$\chi_1 = (\cos(n\varphi), \quad \sin(n\varphi)) \qquad \text{and} \qquad \chi_2 = (-\sin(n\varphi), \quad \cos(n\varphi)) \,. \tag{155}$$

We will show how to find the steerable PDO basis using this example, but the method works exactly the same in all cases. Tables with all explicit solutions can be found in Appendix K.

As described in Section 3.2, we now need to multiply these matrices with the smallest power of $|x|$ such that all entries become polynomials. We show in Appendix J that the necessary coefficient for $\cos(n\varphi)$ and $\sin(n\varphi)$ is $|x|^n$, so in the example above, we get

$$\tilde{\chi}_1 = \left(|x|^n \cos(n\varphi), \quad |x|^n \sin(n\varphi)\right) \qquad \text{and} \qquad \tilde{\chi}_2 = \left(-|x|^n \sin(n\varphi), \quad |x|^n \cos(n\varphi)\right) \,. \tag{156}$$

The entries are written in polar coordinates here, but they are in fact polynomials in the Cartesian coordinates $x_1$ and $x_2$. More precisely, we show in Appendix J that they are closely related to Chebyshev polynomials, based on which we derive the following explicit expressions:

$$|x|^n \cos(n\varphi) = \sum_{i \leq n \text{ even}} (-1)^{\frac{i}{2}} \binom{n}{i} x_1^{n-i} x_2^i \quad \text{and} \quad |x|^n \sin(n\varphi) = \sum_{i \leq n \text{ odd}} (-1)^{\frac{i+1}{2}} \binom{n}{i} x_1^{n-i} x_2^i \,. \tag{157}$$

This Cartesian form then allows us to interpret the polynomial as a differential operator, by applying the ring isomorphism $D$, i.e. plugging in $\partial_1$ and $\partial_2$ for $x_1$ and $x_2$.

If we set $n = 1$ in our example above, which corresponds to a vector field, we get simply $|x| \cos(\varphi) = x_1$ and $|x| \sin(\varphi) = x_2$, so we recover the angular PDO basis

$$D(\tilde{\chi}_1) = (\partial_1, \quad \partial_2) = \text{div} \qquad \text{and} \qquad D(\tilde{\chi}_2) = (-\partial_2, \quad \partial_1) = \text{curl}_{\text{2D}} \tag{158}$$

that we already derived in Section 2.3. To get a complete basis, we combine these PDOs with powers of the Laplacian.

## I.2 Solutions for O(3) and SO(3)

We now turn to the other cases for which steerable kernel solutions have been published, namely $O(3)$ and $SO(3)$. Like for $O(2)$, we only need to consider pairs of irreducible representations. As described in (Weiler et al., 2018a; Lang & Weiler, 2021), we can build the corresponding angular parts $\chi_\beta$ out of real spherical harmonics $Y_{lm}$ using Clebsch-Gordan coefficients. We show in Appendix J that we can apply this procedure to $|x|^l Y_{lm}$ instead of $Y_{lm}$ to obtain the corresponding $\tilde{\chi}_\beta$. We then build the basis by combining with powers of $|x|^2$, as we described in Section 3.2 and already did for $O(2)$. To find the corresponding differential operators $D(\tilde{\chi}_\beta)$, we only need a Cartesian representation of the polynomials $|x|^l Y_{lm}$, which is fortunately well-known (Varshalovich et al., 1988). This then again leads to a complete basis of the space of steerable PDOs, with the same general form containing powers of the Laplacian.

## J Proofs for solutions for subgroups of O(2) and O(3)

In this section, we apply the method from Appendix H to subgroups of $O(2)$ and $O(3)$, by describing bases for the space of steerable kernels that satisfy the conditions of Proposition 19.

### J.1 O(2)

As described in Appendix I.1, it is sufficient to consider irreducible representations $\rho_{\text{in}}$ and $\rho_{\text{out}}$ and for these representations, the angular part of the kernel has a basis consisting of matrices with entries of the form $\cos(n\varphi)$ and $\sin(n\varphi)$ (where $n \in \mathbb{Z}$). Crucially, $n$ is the same for all entries of one matrix (though it may differ between basis elements).

We claim that multiplying each angular basis element with $r^{|n|}$ gives a basis of the space of *polynomial* steerable kernels $\mathcal{K}_{\text{pol}}$ (as a $\mathbb{R}[r^2]$-module), as described in Appendix H. This follows from Proposition 19 if we can prove that $r^{|n|}\cos(n\varphi)$ and $r^{|n|}\sin(n\varphi)$ are

(i) polynomials (in $x, y$),

(ii) homogeneous (of degree $n$),

(iii) and not divisible by $r^2$.

To show that $r^{|n|}\cos(n\varphi)$ and $r^{|n|}\sin(n\varphi)$ are polynomials, we use *Chebyshev polynomials*, which can among other things be seen as a generalization of the addition theorems for $\sin(2\varphi)$ and $\cos(2\varphi)$. Specifically, they are families of polynomials $T_n$ (*first kind*) and $U_n$ (*second kind*) defined for $n \geq 0$ with the property that

$$\cos(n\varphi) = T_n(\cos\varphi) \tag{159}$$
$$\sin(n\varphi) = U_{n-1}(\cos\varphi)\sin\varphi\,. \tag{160}$$

We extend the definition to negative $n$ by setting

$$T_{-n} := T_n \qquad\qquad \text{for } n \geq 0 \tag{161}$$
$$U_{-1} := 0 \tag{162}$$
$$U_{-n-1} := -U_{n-1} \qquad\qquad \text{for } n \geq 1\,. \tag{163}$$

Then Eq. (159) holds for all $n \in \mathbb{Z}$, as follows immediately from the parity of $\sin$ and $\cos$.

Motivated by the close relation to Chebyshev polynomials, we then define the following notation for the matrix entries we are considering:

$$\tilde{T}_n := r^{|n|}T_n(\cos\varphi) = r^{|n|}\cos(n\varphi) \tag{164}$$
$$\tilde{U}_n := r^{|n|}\sin(\varphi)U_{n-1}(\cos\varphi) = r^{|n|}\sin(n\varphi)\,. \tag{165}$$

We now prove claim (i), that $\tilde{T}_n$ and $\tilde{U}_n$ are polynomials in the Cartesian coordinates $x, y$, by using a few well-known facts about Chebyshev polynomials: First, $T_n$ has degree $|n|$, so the highest order term in $\tilde{T}_n$ is $r^{|n|}(\cos\varphi)^{|n|} = x^{|n|}$. Similarly, $U_{n-1}$ has degree $|n| - 1$ and the highest order term

in $\tilde{U}_n$ is thus $r^{|n|}(\cos\varphi)^{|n|-1}\sin\varphi = x^{|n|-1}y$. Lower order terms have additional powers of $r$ that aren't "matched" by a cosine or sine. But the second fact about Chebyshev polynomials is that they are either even or odd, so all of these powers of $r$ are even (i.e. powers of $r^2$) and thus themselves polynomials.

That $\tilde{T}_n$ and $\tilde{U}_n$ are homogeneous of degree $n$ – claim (ii) – is immediately clear from their definition.

It remains to show claim (iii), that they are not divisible by $r^2$. For that, we use the following Lemma:

**Lemma 23.** *Let $p$ be a non-zero, harmonic, homogeneous polynomial. Then $p$ is not divisible by $r^2$.*

Here, $p$ is harmonic if its Laplacian vanishes.

*Proof.* For any homogeneous polynomial $p$, there is a *unique* decomposition

$$p = r^2 q + h\,, \tag{166}$$

such that $h$ is harmonic and homogeneous. Since $q = 0, h = p$ is one such decomposition for $p$ harmonic, there is no solution with $h = 0$ (unless $p = 0$). So non-zero homogeneous harmonic polynomials are not divisible by $r^2$. $\qquad\square$

Now we only need to show that $\tilde{T}_n$ and $\tilde{U}_n$ are in fact harmonic. This can be done with a brief calculation in polar coordinates:

$$\begin{aligned}
\Delta\tilde{T}_n &= \left(\partial_r^2 + \frac{1}{r}\partial_r + \frac{1}{r^2}\partial_\varphi^2\right)(r^n\cos(n\varphi))\\
&= n(n-1)r^{n-2}\cos(n\varphi) + nr^{n-2}\cos(n\varphi) - n^2r^{n-2}\cos(n\varphi)\\
&= 0\,.
\end{aligned} \tag{167}$$

The same holds for $\tilde{U}_n$:

$$\begin{aligned}
\Delta\tilde{U}_n &= \left(\partial_r^2 + \frac{1}{r}\partial_r + \frac{1}{r^2}\partial_\varphi^2\right)(r^n\sin(n\varphi))\\
&= n(n-1)r^{n-2}\sin(n\varphi) + nr^{n-2}\sin(n\varphi) - n^2r^{n-2}\sin(n\varphi)\\
&= 0\,.
\end{aligned} \tag{168}$$

In summary, we have shown that $\tilde{T}_n$ and $\tilde{U}_n$ are homogeneous polynomials not divisible by $r^2$, which makes Proposition 19 applicable.

In order to make the basis practically applicable, we also give explicit Cartesian expressions for $\tilde{T}_n$ and $\tilde{U}_n$. We restrict ourselves to non-negative $n$ to keep the notation less cluttered. Equation (161) immediately gives the case for $n < 0$. An explicit formula for the Chebyshev polynomials of the first kind is

$$T_n(x) = \sum_{k=0}^{\lfloor\frac{n}{2}\rfloor}\binom{n}{2k}\left(x^2-1\right)^k x^{n-2k}\,. \tag{169}$$

It follows that

$$\begin{aligned}
\tilde{T}_n &= r^n T_n(\cos\varphi)\\
&= \sqrt{x^2+y^2}^n T_n\left(\frac{x}{\sqrt{x^2+y^2}}\right)\\
&= \sum_{k=0}^{\lfloor\frac{n}{2}\rfloor}\binom{n}{2k}\left(-y^2\right)^k x^{n-2k}\\
&= \sum_{k=0}^{\lfloor\frac{n}{2}\rfloor}\binom{n}{2k}(-1)^k y^{2k}x^{n-2k}\,.
\end{aligned} \tag{170}$$

Similarly, the Chebyshev polynomials of the second kind are given by

$$U_n(x) = \sum_{k=0}^{\lfloor \frac{n}{2} \rfloor} \binom{n+1}{2k+1} \left(x^2 - 1\right)^k x^{n-2k}, \tag{171}$$

which means that

$$
\begin{aligned}
\tilde{U}_n &= r^n \sin\varphi \, U_{n-1}(\cos\varphi) \\
&= y \sqrt{x^2 + y^2}^{\,n-1} U_{n-1}\left(\frac{x}{\sqrt{x^2 + y^2}}\right) \\
&= y \sum_{k=0}^{\lfloor \frac{n-1}{2} \rfloor} \binom{n}{2k+1} \left(-y^2\right)^k x^{n-1-2k} \\
&= \sum_{k=0}^{\lfloor \frac{n-1}{2} \rfloor} \binom{n}{2k+1} (-1)^k y^{2k+1} x^{n-(2k+1)}.
\end{aligned} \tag{172}
$$

## J.2   O(3)

As already mentioned in Appendix I.2, for $O(3)$ and $SO(3)$, the irreps angular basis between irreducible representation can be built by combining real spherical harmonics $Y_{lm}$ using Clebsch-Gordan coefficients. We refer to Weiler et al. (2018a); Lang & Weiler (2021) for details on how this works since it is exactly the same procedure whether one uses kernels or PDOs. The only fact we need to know here is that each basis element is built from spherical harmonics with the same $l$.

The necessary steps are now very similar to those for $O(2)$ and its subgroups. We will use $r^l Y_{lm}$ where we had $\tilde{T}_n$ and $\tilde{U}_n$ for $O(2)$. Here it becomes important that inside one basis element, only one $l$ appears, because this means we can multiply the entire matrix of polynomials by $r^l$.

We would then need to show that these are homogeneous polynomials not divisible by $r^2$ but for spherical harmonics, it is already very well known that $r^l Y_{lm}$ are homogeneous harmonic polynomials of degree $l$; Lemma 23 then implies that they are not divisible by $r^2$. For explicit Cartesian expressions, we refer to e.g. Varshalovich et al. (1988).

## K   SOLUTION TABLES

Using the results from Appendices H and J, we now very easily get complete bases for all subgroups of $O(2)$ and for all irreducible representations by transferring the solutions by Weiler & Cesa (2019). The appendix of (Weiler & Cesa, 2019) contains tables with all *kernel* solutions; we simply replace every term of the form $\cos(k\varphi)$ with $\tilde{T}_k$ and $\sin(k\varphi)$ with $\tilde{U}_k$ to get the following tables.

For the sake of readability, we only write the polynomials $\tilde{T}_k$ and $\tilde{U}_k$ inside the tables, though the PDOs themselves should of course be $D(\tilde{T}_k)$ and $D(\tilde{U}_k)$. Explicit formulas for $\tilde{T}_k$ and $\tilde{U}_k$ were given in Eq. (157).

All bases described here are *module* bases for the space of steerable PDOs as an $\mathbb{R}[\Delta]$-module. See Section 3.2 and Appendix H for details.

### K.1   SPECIAL ORTHOGONAL GROUP $SO(2)$

The irreducible representations of $SO(2)$ are the trivial representation $\psi_0$ and those of the form

$$\psi_k : SO(2) \to GL(\mathbb{R}^2), \qquad g \mapsto g^k. \tag{173}$$

The bases for all the combinations of these irreps are:

| Out \ In | $\psi_0$ | $\psi_n, \quad n \in \mathbb{N}_{>0}$ |
|---|---|---|
| $\psi_0$ | (1) | $\begin{pmatrix} \tilde{T}_n & \tilde{U}_n \end{pmatrix}, \begin{pmatrix} -\tilde{U}_n & \tilde{T}_n \end{pmatrix}$ |
| $\psi_m,$ $m \in \mathbb{N}_{>0}$ | $\begin{pmatrix} \tilde{T}_m \\ \tilde{U}_m \end{pmatrix}, \begin{pmatrix} -\tilde{U}_m \\ \tilde{T}_m \end{pmatrix}$ | $\begin{pmatrix} \tilde{T}_{m-n} & -\tilde{U}_{m-n} \\ \tilde{U}_{m-n} & \tilde{T}_{m-n} \end{pmatrix}, \begin{pmatrix} -\tilde{U}_{m-n} & -\tilde{T}_{m-n} \\ \tilde{T}_{m-n} & -\tilde{U}_{m-n} \end{pmatrix},$ $\begin{pmatrix} \tilde{T}_{m+n} & \tilde{U}_{m+n} \\ \tilde{U}_{m+n} & -\tilde{T}_{m+n} \end{pmatrix}, \begin{pmatrix} -\tilde{U}_{m+n} & \tilde{T}_{m+n} \\ \tilde{T}_{m+n} & \tilde{U}_{m+n} \end{pmatrix}$ |

## K.2 Orthogonal group O(2)

Elements of $O(2)$ can be written as a tuple $(r, s)$ consisting of a rotation $r \in SO(2)$ and a flip $s \in \{\pm 1\}$. The irreducible representations with frequency $k > 0$ are similar to those for $SO(2)$, only with the additional flip:

$$\psi_{1,k} : SO(2) \to GL(\mathbb{R}^2), \qquad (r, s) \mapsto r^k \circ s. \tag{174}$$

Here, $s$ is understood as a map acting on $\mathbb{R}^2$ either as the identity or by flipping along a certain axis. In contrast to $SO(2)$, there are now two irreducible representations for $k = 0$, namely the trivial representation $\psi_{0,0}(r, s) = 1$ and the representation $\psi_{1,0}(r, s) = s$. The solutions for all possible combinations of these irreducible representations are as follows:

| Out \ In | $\psi_{0,0}$ | $\psi_{1,0}$ | $\psi_{1,n}, \quad n \in \mathbb{N}_{>0}$ |
|---|---|---|---|
| $\psi_{0,0}$ | (1) | $\emptyset$ | $\begin{pmatrix} -\tilde{U}_n & \tilde{T}_n \end{pmatrix}$ |
| $\psi_{1,0}$ | $\emptyset$ | (1) | $\begin{pmatrix} \tilde{T}_n & \tilde{U}_n \end{pmatrix}$ |
| $\psi_{1,m},$ $m \in \mathbb{N}_{>0}$ | $\begin{pmatrix} -\tilde{U}_m \\ \tilde{T}_m \end{pmatrix}$ | $\begin{pmatrix} \tilde{T}_m \\ \tilde{U}_m \end{pmatrix}$ | $\begin{pmatrix} \tilde{T}_{m-n} & -\tilde{U}_{m-n} \\ \tilde{U}_{m-n} & \tilde{T}_{m-n} \end{pmatrix}, \begin{pmatrix} \tilde{T}_{m+n} & \tilde{U}_{m+n} \\ \tilde{U}_{m+n} & -\tilde{T}_{m+n} \end{pmatrix}$ |

## K.3 Reflection group ±1

The reflection group has only two irreducible representations: the trivial representation $\psi_0$ and the representation $\psi_1(s) = s$. Both are one-dimensional, so all the PDOs are only $1 \times 1$ matrices:

| Out \ In | $\psi_0$ | $\psi_1$ |
|---|---|---|
| $\psi_0$ | $\begin{pmatrix} \tilde{T}_\mu \end{pmatrix}$ | $\begin{pmatrix} \tilde{U}_\mu \end{pmatrix}$ |
| $\psi_1$ | $\begin{pmatrix} \tilde{U}_\mu \end{pmatrix}$ | $\begin{pmatrix} \tilde{T}_\mu \end{pmatrix}$ |

To get the full basis, $\mu$ needs to take on all natural numbers (in practice, we use all $\mu$ up to some maximum value). Note that we assume that the reflection is with respect to the $x_1$-axis, both here and later for $D_N$. To get other reflection axes, the PDOs simply need to be rotated.

## K.4 Cyclic group $C_N$

The irreducible representations of $C_N$ are the same as those of $SO(2)$ but only up to a frequency of $k = \lfloor \frac{N-1}{2} \rfloor$. If $N$ is even, there is an additional one-dimensional irreducible representation, namely

$$\psi_{N/2}(\theta) = \cos\left(\frac{N}{2}\theta\right). \tag{175}$$

We then get the following solutions, where $t$ ranges over $\mathbb{Z}$ and $\hat{t}$ over $\mathbb{N}$:

| In / Out | $\psi_0$ | $\psi_{N/2}$ (if $N$ even) | $\psi_n$, $1 \le n < N/2$ |
|---|---|---|---|
| $\psi_0$ | $\left(\tilde{T}_{\hat{t}N}\right)$, $\left(\tilde{U}_{\hat{t}N}\right)$ | $\left(\tilde{T}_{(\hat{t}+1/2)N}\right)$, $\left(\tilde{U}_{(\hat{t}+1/2)N}\right)$ | $\left(-\tilde{U}_{n+tN} \quad \tilde{T}_{n+tN}\right)$, $\left(\tilde{T}_{n+tN} \quad \tilde{U}_{n+tN}\right)$ |
| $\psi_1$ ($N$ even) | $\left(\tilde{T}_{(\hat{t}+1/2)N}\right)$, $\left(\tilde{U}_{(\hat{t}+1/2)N}\right)$ | $\left(\tilde{T}_{\hat{t}N}\right)$, $\left(\tilde{U}_{\hat{t}N}\right)$ | $\left(-\tilde{U}_{n+(t+1/2)N} \quad \tilde{T}_{n+(t+1/2)N}\right)$, $\left(\tilde{T}_{n+(t+1/2)N} \quad \tilde{U}_{n+(t+1/2)N}\right)$ |
| $\psi_n$ $1 \le n < N/2$ | $\left(\begin{smallmatrix}-\tilde{U}_{m+tN}\\ \tilde{T}_{m+tN}\end{smallmatrix}\right)$, $\left(\begin{smallmatrix}\tilde{T}_{m+tN}\\ \tilde{U}_{m+tN}\end{smallmatrix}\right)$ | $\left(\begin{smallmatrix}-\tilde{U}_{m+(t+1/2)N}\\ \tilde{T}_{m+(t+1/2)N}\end{smallmatrix}\right)$, $\left(\begin{smallmatrix}\tilde{T}_{m+(t+1/2)N}\\ \tilde{U}_{m+(t+1/2)N}\end{smallmatrix}\right)$ | $\left(\begin{smallmatrix}\tilde{T}_{m-n+tN} & -\tilde{U}_{m-n+tN}\\ \tilde{U}_{m-n+tN} & \tilde{T}_{m-n+tN}\end{smallmatrix}\right)$, $\left(\begin{smallmatrix}\tilde{T}_{m+n+tN} & \tilde{U}_{m+n+tN}\\ \tilde{U}_{m+n+tN} & -\tilde{T}_{m+n+tN}\end{smallmatrix}\right)$, $\left(\begin{smallmatrix}-\tilde{U}_{m-n+tN} & -\tilde{T}_{m-n+tN}\\ \tilde{T}_{m-n+tN} & -\tilde{U}_{m-n+tN}\end{smallmatrix}\right)$, $\left(\begin{smallmatrix}-\tilde{U}_{m+n+tN} & \tilde{T}_{m+n+tN}\\ \tilde{T}_{m+n+tN} & \tilde{U}_{m+n+tN}\end{smallmatrix}\right)$ |

## K.5 Dihedral Group $D_N$

Similarly to $C_N$, the irreducible representations of $D_N$ are the same as those of O(2) up to a frequency of $k = \lfloor \frac{N-1}{2} \rfloor$. If $N$ is even, there are two additional one-dimensional irreducible representations, namely

$$\psi_{0,N/2}(\theta, s) = \cos\left(\frac{N}{2}\theta\right),$$
$$\psi_{1,N/2}(\theta, s) = s\cos\left(\frac{N}{2}\theta\right). \tag{176}$$

The solutions are (again with $t \in \mathbb{Z}$ and $\hat{t} \in \mathbb{N}$):

| In / Out | $\psi_{0,0}$ | $\psi_{1,0}$ | $\psi_{0,N/2}$ ($N$ even) | $\psi_{1,N/2}$ ($N$ even) | $\psi_{1,n}$, $1 \le n < N/2$ |
|---|---|---|---|---|---|
| $\psi_{0,0}$ | $\left(\tilde{T}_{\hat{t}N}\right)$ | $\left(\tilde{U}_{\hat{t}N}\right)$ | $\left(\tilde{T}_{(\hat{t}+1/2)N}\right)$ | $\left(\tilde{U}_{(\hat{t}+1/2)N}\right)$ | $\left(-\tilde{U}_{n+tN} \quad \tilde{T}_{n+tN}\right)$ |
| $\psi_{1,0}$ | $\left(\tilde{U}_{\hat{t}N}\right)$ | $\left(\tilde{T}_{\hat{t}N}\right)$ | $\left(\tilde{U}_{(\hat{t}+1/2)N}\right)$ | $\left(\tilde{T}_{(\hat{t}+1/2)N}\right)$ | $\left(\tilde{T}_{n+tN} \quad \tilde{U}_{n+tN}\right)$ |
| $\psi_{0,N/2}$, ($N$ even) | $\left(\tilde{T}_{(\hat{t}+1/2)N}\right)$ | $\left(\tilde{U}_{(\hat{t}+1/2)N}\right)$ | $\left(\tilde{T}_{\hat{t}N}\right)$ | $\left(\tilde{U}_{\hat{t}N}\right)$ | $\left(-\tilde{U}_{n+(t+1/2)N} \quad \tilde{T}_{n+(t+1/2)N}\right)$ |
| $\psi_{1,N/2}$, ($N$ even) | $\left(\tilde{U}_{(\hat{t}+1/2)N}\right)$ | $\left(\tilde{T}_{(\hat{t}+1/2)N}\right)$ | $\left(\tilde{U}_{\hat{t}N}\right)$ | $\left(\tilde{T}_{\hat{t}N}\right)$ | $\left(\tilde{T}_{n+(t+1/2)N} \quad \tilde{U}_{n+(t+1/2)N}\right)$ |
| $\psi_{1,m}$, $1 \le m \le N/2$ | $\left(\begin{smallmatrix}-\tilde{U}_{m+tN}\\ \tilde{T}_{m+tN}\end{smallmatrix}\right)$ | $\left(\begin{smallmatrix}\tilde{T}_{m+tN}\\ \tilde{U}_{m+tN}\end{smallmatrix}\right)$ | $\left(\begin{smallmatrix}-\tilde{U}_{m+(t+1/2)N}\\ \tilde{T}_{m+(t+1/2)N}\end{smallmatrix}\right)$ | $\left(\begin{smallmatrix}\tilde{T}_{m+(t+1/2)N}\\ \tilde{U}_{m+(t+1/2)N}\end{smallmatrix}\right)$ | $\left(\begin{smallmatrix}\tilde{T}_{m-n+tN} & -\tilde{U}_{m-n+tN}\\ \tilde{U}_{m-n+tN} & \tilde{T}_{m-n+tN}\end{smallmatrix}\right)$, $\left(\begin{smallmatrix}\tilde{T}_{m+n+tN} & \tilde{U}_{m+n+tN}\\ \tilde{U}_{m+n+tN} & -\tilde{T}_{m+n+tN}\end{smallmatrix}\right)$ |

## L  Discretization Methods for PDOs

### L.1  Finite differences

Finite difference methods are common in machine learning; for example, the discretization of $\frac{d}{dx}$ as $\begin{bmatrix}-1 & 1\end{bmatrix}$ or $\begin{bmatrix}-1 & 0 & 1\end{bmatrix}$, or of $\frac{d^2}{dx^2}$ as $\begin{bmatrix}1 & -2 & 1\end{bmatrix}$ all use finite difference methods. To understand where these filters come from, we need the following well-known result:

**Proposition 24.** *Let $x_1, \ldots, x_N \in \mathbb{R}$ be arbitrary but distinct grid points. Then for $m \le N - 1$, there are unique coefficients $w_n^{(m)}$ such that the approximation*

$$f^{(m)}(0) \approx \frac{1}{h^m} \sum_{n=1}^{N} w_n^{(m)} f(hx_n) \tag{177}$$

*has an error $\mathcal{O}(h^{N-m})$ for any $f \in C^N(\mathbb{R})$.*

The coefficients $w_n^{(m)}$ are called *finite difference coefficients* and approximating derivatives using Eq. (177) is the finite difference method. We will soon describe how to generalize this to higher dimensions as well.

We remark that $\mathcal{O}(h^{N-m})$ is an asymptotic *upper bound* on the error, and it can sometimes be lower, even for all $f$. For example, the central difference discretization of $\frac{d^2}{dx^2}$ as $\begin{bmatrix} 1 & -2 & 1 \end{bmatrix}$ uses $N = 3$ grid points but still achieves an error of $\mathcal{O}(h^2)$, rather than $\mathcal{O}(h)$. For details on when such a "boosted" order of accuracy occurs, see (Sadiq & Viswanath, 2014).

Note that Eq. (177) can be generalized to

$$f^{(m)}(x) \approx \frac{1}{h^m} \sum_{n=1}^{N} w_n^{(m)} f(hx_n + x) \,. \tag{178}$$

This follow immediately because the coefficients $w_n^{(m)}$ don't depend on the function $f$, so we can apply Proposition 24 to $\tau_{-x} f$.

Particularly interesting for us is the case of a regular grid. We can use infinitely many grid points $x_n \in \mathbb{Z}$ as long as we demand that $w_n^{(m)}$ is zero for almost all $n$. Then we get

$$f^{(m)}(x) \approx \frac{1}{h^m} \sum_{n} w_n^{(m)} f(hn + x) \,. \tag{179}$$

Fixing $h = 1$, this is exactly the cross correlation

$$f^{(m)} \approx w^{(m)} \star f \tag{180}$$

if we interpret $w^{(m)}$ as a function $n \mapsto w_n^{(m)}$. This is why, in the end, we discretize a derivative by convolving with some stencil, such as $\begin{bmatrix} 1 & -2 & 1 \end{bmatrix}$, at least on a regular 1D grid.

Generalizing Proposition 24 to higher dimensions does not work in a straightforward way. However, if we restrict ourselves to regular grids, then finite difference methods can be easily applied to PDOs. The idea is very simple: a PDO such as $\partial_x \partial_y^2$ can be interpreted as first applying $\partial_y^2$ and then $\partial_x$ (or the other way around). So we discretize each of these with the one-dimensional finite difference method described before, and then we convolve with both filters one after the other.[8] We can also combine the two one-dimensional filter into one two-dimensional filter, the outer product of the two.[9] The asymptotic error of this discretization will simply be the highest asymptotic error along all the dimensions, so we get similar guarantees.

### L.2 RBF-FD

As mentioned, Proposition 24 does not directly generalize to higher dimensions. So to discretize a PDO on arbitrary point clouds in higher dimensions, a somewhat different approach is needed.

RBF-FD is one such method and works as follows: we still want to approximate a derivative using

$$\partial^\alpha f(0) \approx \sum_{n=1}^{N} w_n^\alpha f(x_n) \,, \tag{181}$$

similar to finite difference methods. Here, $x_n \in \mathbb{R}^d$ are arbitrary (but again distinct) points. The idea is now that we require this approximation to be exact if $f(x) = \varphi(\|x - x_n\|)$, where $\varphi$ is an arbitrary but fixed radial basis function. In words, the approximation should become exact for a certain radial basis function centered on any of the points $x_n$. This leads to a linear system, which is solved for the coefficients $w_n^\alpha$.[10] For more details on both finite differences and RBF-FD, see for example Fornberg & Flyer (2015).

---

[8]As we have seen, finite difference methods can most immediately be seen as performing a cross-correlation rather than a convolution. However, we can easily switch to convolutions by flipping the filter.

[9]For a simple PDO such as $\partial_x \partial_y^2$, this may be undesirable for computational reasons. But in practice, we have PDOs that are sums of such pure terms and thus don't factorize.

[10]In practice, one often solves an extended linear system containing additional low-order polynomials, but we won't discuss that here.

### L.3 Gaussian derivatives

Discretizing PDOs using derivatives of Gaussians is very simple to describe: given grid points $x_n \in \mathbb{R}^d$, we approximate using

$$\partial^\alpha f(0) \approx \sum_{n=1}^{N} \left( \partial^\alpha G(x_n; \sigma) \right) f(x_n) \tag{182}$$

where $G(x; \sigma)$ is a Gaussian kernel with standard deviation $\sigma$ centered around 0. $\sigma$ is a free parameter; larger $\sigma$ will lead to a stronger denoising effect.

On regular grids, this again turns into a cross-correlation, with the filter being the derivative $\partial^\alpha G(x_n; \sigma)$ evaluated on the grid coordinates.

In Appendix E we briefly touch on a possible interpretation of this discretization method using the distributional framework for PDOs.

## M    Relation to PDO-eConvs

In this section, we describe how PDO-eConvs (Shen et al., 2020) fit into the framework of steerable PDOs. We mostly follow the original notation from Shen et al. (2020) when describing PDO-eConvs but do make some minor changes to avoid clashes and confusion with our own notation.

As in our presentation, Shen et al. (2020) use polynomials to describe PDOs. One difference is that they never explicitly use *matrices* of polynomials, because they model the feature space somewhat differently (which we will discuss in a moment). They write $H$ for the polynomial describing a PDO (where we would write e.g. $p$) and write $\chi^{(A)}$ for the corresponding PDO transformed by $A \in \mathrm{O}(d)$. In our notation,

$$\chi^{(A)} := D(A \cdot H) = D(H \circ A^{-1}) \,. \tag{183}$$

PDO-eConvs use two types of PDO layers. The first one, $\Psi$, can be interpreted as a steerable PDO with $\rho_{\mathrm{in}}$ trivial and $\rho_{\mathrm{out}}$ regular. It maps the scalar input to the network to the internally used regular representation. The second layer type, $\Phi$ maps between regular representations and is used for hidden layers. At the end, pooling is performed to obtain a scalar output again.

The first layer type is defined as

$$\Psi : C^\infty(\mathbb{R}^d) \to C^\infty(\tilde{E}(d)), \qquad \Psi(f)(x, A) := (\chi^{(A)} f)(x) \,. \tag{184}$$

Here, $\tilde{E}(d) := \mathbb{R}^d \rtimes S$ with $S \leq \mathrm{O}(d)$; in practice, $S$ needs to be a finite subgroup, i.e. $C_N$ or $D_N$. Elements of $\tilde{E}(d)$ can be uniquely written as $(x, A)$ with $x \in \mathbb{R}^d$ and $A \in S$.

There is an obvious bijection $C^\infty(\tilde{E}(d)) \cong C^\infty(\mathbb{R}^d, \mathbb{R}^c)$, where $c := |S|$ is the order of $S$, i.e. the number of group elements. Concretely, we define

$$\Theta : C^\infty\left(\tilde{E}(d)\right) \to C^\infty(\mathbb{R}^d, \mathbb{R}^c), \qquad f \mapsto \left(x \mapsto \left(f(x, A_1), \ldots, f(x, A_c)\right)\right) \tag{185}$$

where $A_1, \ldots, A_c$ is an enumeration of the group elements of $S$. We will therefore interpret $C^\infty(\tilde{E}(d))$ as a $c$-dimensional feature field over $\mathbb{R}^d$, and we will show that using regular representations for this field (and trivial ones for the input) makes the PDO-eConv layers equivariant and thus steerable PDOs.

First, note that under the $\Theta$ bijection, the first PDO-eConv layer type $\Psi$ becomes a $c \times 1$ matrix of PDOs, namely

$$D(H_\Psi) := D\left( \begin{pmatrix} A_1 \cdot H \\ \vdots \\ A_c \cdot H \end{pmatrix} \right) = \begin{pmatrix} \chi^{(A_1)} \\ \vdots \\ \chi^{(A_c)} \end{pmatrix} \,. \tag{186}$$

What we mean by this is that the diagram

$$C^\infty(\mathbb{R}^d) \xrightarrow{\quad \Psi \quad} C^\infty(\tilde{E}(d))$$
$$\| \qquad\qquad \downarrow \Theta$$
$$C^\infty(\mathbb{R}^d) \xrightarrow{\quad D(H_\Psi) \quad} C^\infty(\mathbb{R}^d, \mathbb{R}^c)$$

commutes. Concretely, we have

$$\Psi(f)(x, A_i) = (\chi^{(A_i)}f)(x) = (D(H_\Psi)f)(x)_i. \tag{187}$$

So we need to check whether $H_\Psi$ satisfies the PDO steerability constraint for trivial to regular PDOs:

$$H_\Psi(Ax) = \rho_{\text{regular}}(A)H_\Psi(x). \tag{188}$$

Using the definition of $\rho_{\text{regular}}(A)$, we can rewrite the RHS as

$$\begin{aligned}
\rho_{\text{regular}}(A)H_\Psi(x) &= \sum_{k=1}^c \rho_{\text{regular}}(A)H_\psi(x)_k e_{A_k} \\
&= \sum_{k=1}^c (A_k \cdot H)(x)e_{AA_k} \\
&= \sum_{k=1}^c H(A_k^{-1}x)e_{AA_k} \\
&= \sum_{l=1}^c H(A_l^{-1}Ax)e_{A_l}.
\end{aligned} \tag{189}$$

Here, we use basis vectors $e_{A_1}, \ldots, e_{A_c}$ for $\mathbb{R}^c$, with the same enumeration $A_1, \ldots, A_c$ of $S$ used to define $\Theta$. For the final step, we reparameterized the sum with $A_l := AA_k$.

The LHS of Eq. (188) can be written as

$$\begin{aligned}
H_\Psi(Ax) &= (A_1 \cdot H(Ax), \quad \ldots, \quad A_c \cdot H(Ax))^T \\
&= (H(A_1^{-1}Ax), \quad \ldots, \quad H(A_c^{-1}Ax))^T.
\end{aligned} \tag{190}$$

This is the same as the final term in Eq. (189), which proves that the PDO steerability constraint is satisfied.

The second PDO-eConv layer type, mapping between regular representations, is defined as

$$\Phi : C^\infty(\tilde{E}(d)) \to C^\infty(\tilde{E}(d)), \qquad \Phi(e)(x, A) := \sum_{j=1}^c \left(\chi_{A_j}^{(A)}e\right)(x, AA_j). \tag{191}$$

$\chi_{A_j}^{(A)}$ are $c$ different PDOs and they act on $e \in C^\infty(\tilde{E}(d))$ by acting on each of the $c$ components separately. Under the $\Theta$ bijection, $\Phi$ becomes

$$\begin{aligned}
\Theta(\Phi(e))(x)_i &= \sum_{j=1}^c \left(\chi_{A_j}^{(A_i)}e\right)(x, A_iA_j) \\
&= \sum_{j=1}^c \left(\chi_{A_j}^{(A_i)}\Theta(e)_{A_iA_j}\right)(x) \\
&= \sum_{j=1}^c \left(\chi_{A_i^{-1}A_j}^{(A_i)}\Theta(e)_{A_j}\right)(x).
\end{aligned} \tag{192}$$

We can therefore represent it as a $c \times c$ PDO $D(H_\Phi)$ with

$$(H_\Phi)_{ij}(x) = H_{A_i^{-1}A_j}(A_i^{-1}x) =: H_{ij}(x), \tag{193}$$

where $H_{A_i^{-1}A_j}$ is the polynomial that induces $\chi_{A_i^{-1}A_j}^{(I)}$. This makes the diagram

$$
\begin{array}{ccc}
C^\infty(\tilde{E}(d)) & \xrightarrow{\ \Phi\ } & C^\infty(\tilde{E}(d)) \\
\downarrow{\scriptstyle\Theta} & & \downarrow{\scriptstyle\Theta} \\
C^\infty(\mathbb{R}^d, \mathbb{R}^c) & \xrightarrow{\ D(H_\Phi)\ } & C^\infty(\mathbb{R}^d, \mathbb{R}^c)
\end{array}
$$

commute, similar to the case discussed above. So again, we need to check that $H_\Phi$ satisfies the PDO steerability constraint, this time for $\rho_{\text{in}}$ and $\rho_{\text{out}}$ both regular:

$$
H_\Phi(Ax) = \rho_{\text{regular}}(A) H_\Phi(x) \rho_{\text{regular}}(A^{-1}) . \tag{194}
$$

Writing out $H_\Psi$ in its components, this becomes

$$
\begin{aligned}
H_\Psi(Ax) &\overset{!}{=} \sum_{i,j} \rho_{\text{regular}}(A) H_{ij}(x) e_{A_i} e_{A_j}^T \rho_{\text{regular}}(A)^{-1} \\
&\overset{(1)}{=} \sum_{i,j} H_{ij}(x) e_{AA_i} e_{AA_j}^T \\
&= \sum_{i,j} H_{A_i^{-1} A_j}(A_i^{-1}x) e_{AA_i} e_{AA_j}^T \\
&\overset{(2)}{=} \sum_{i,j} H_{A_i^{-1} A A^{-1} A_j}(A_i^{-1} Ax) e_{A_i} e_{A_j}^T \\
&= \sum_{i,j} H_{A_i^{-1} A_j}(A_i^{-1} Ax) e_{A_i} e_{A_j}^T \\
&= \sum_{i,j} H_{ij}(Ax) e_{A_i} e_{A_j}^T .
\end{aligned} \tag{195}
$$

The first and the last term are the same, just written out in components on the RHS, so the steerability constraint is again satisfied. For (1), we used that $\rho_{\text{regular}}(A)$ is orthogonal, and thus

$$
e_j^T \rho_{\text{regular}}(A)^{-1} = e_j^T \rho_{\text{regular}}(A)^T = \left(\rho_{\text{regular}}(A) e_j\right)^T . \tag{196}
$$

(2) was again a reparameterization of the sum, with $AA_i \mapsto A_i$ and $AA_j \mapsto A_j$. The other steps are only simplifications and plugging in definitions.

In conclusion, we have shown that there is a simple bijection between the feature spaces used for PDO-eConv hidden layers and the feature fields we use, and that under this bijection, PDO-eConvs correspond to steerable PDOs with regular representations (and trivial representations for the input).

It is relatively easy to adapt the argument we present for the converse direction: every equivariant PDO between two regular feature fields (or from a scalar to a regular one) can be interpreted as a PDO-eConv layer.

# N    ADDITIONAL EXPERIMENTAL RESULTS

## N.1    FLUID FLOW PREDICTION

While our classification experiments on MNIST-rot and STL-10 make use of various representations inside the networks, the input and output are always trivial. To showcase the flexibility of the steerable PDO framework, we also conduct a small experiment in fluid flow prediction, where the output contains a vector field.

We use the dataset provided by Ribeiro et al. (2020). It contains simulated laminar fluid flows around various objects. The inputs are all scalar fields describing the geometry of the problem. The outputs are the velocity and pressure field, the former of which is a vector field, and thus requires vector representations to get the correct equivariance (rather than only trivial and regular ones like PDO-eConvs support). In the original dataset, the fluid always flows into a tube from the same direction. To better demonstrate the effects of equivariance, we randomly rotate each sample by a multiple of $\frac{\pi}{2}$, making the task somewhat more challenging. During training, the rotations are

Table 3: Mean squared test error for prediction of velocity and pressure of laminar flow around different objects.

| Method | Equivariance | MSE |
|--------|--------------|-----|
| Kernel | — | $3.20 \pm 0.22$ |
|        | $C_8$ | $2.26 \pm 0.14$ |
| PDO | — | $2.75 \pm 0.21$ |
|     | $C_8$ | $2.32 \pm 0.08$ |

different in each epoch to avoid disadvantaging the non-equivariant networks (i.e. we effectively use data augmentation). To ensure all inputs have the same shape, we pad the original samples with zeros to make them square before rotating them.

Our architecture and hyperparameters follow those of Ribeiro et al. (2020). The only exception is that we use a single decoder for the vector field (since we treat it as a single vector-valued output for the purposes of equivariance), whereas Ribeiro et al. (2020) used separate decoders for the two vector components in some of their experiments. Just as in the STL-10 experiments, we perform no hyperparameter tuning and use the hyperparameters that Ribeiro et al. (2020) optimized for the non-equivariant network. To make the network equivariant, we replace the usual convolutions with $C_8$-equivariant steerable kernels or PDOs. We chose the channel sizes such that all networks had approximately the same number of parameters (slightly over 800k).

Table 3 shows a clear advantage of the equivariant networks over the non-equivariant ones.[11] Steerable PDOs perform slightly worse than steerable kernels, though the difference is within the error intervals. They still perform clearly better than any non-equivariant method. The PDO results are based on Gaussian discretization, since that performed best in our other experiments.

## N.2 EQUIVARIANCE ERRORS

To check the equivariance error—and indirectly the discretization error, since at least equivariant layers have zero equivariance error in the continuous setting—, we checked how much rotating an input image changes the output of a layer, compared to what the output should be under perfect equivariance. The challenge here is that rotating a discrete image itself introduces some errors. To minimize those, we used a large high-dimensional image, rotated it, and then scaled it down before passing it into the layer, and scaled down again after that. We compared the result of this procedure to what we get by first downscaling, then applying a convolutional or PDO layer, then rotating, and then downscaling again. Effectively, all rotations thus happen at large resolutions, which should minimize artifacts.

Table 4 shows the relative equivariance errors (as multiples of 1e-6). These errors are for randomly initialized layers (averaged over 10 initializations). As discussed in the main text, the asymptotic error bound for finite difference discretization does not lead to a particularly low error in practice.

## N.3 RESTRICTION EXPERIMENTS

Table 5 shows additional results on MNIST-rot. The general architecture and hyperparameters are the same as in the experiments in Section 4 with regular representations, using our basis. However, in the experiments in this section, the first five layers are $D_{16}$-equivariant, while the final PDO/convolutional layer is $C_{16}$-equivariant. The motivation for this is that while the input images do not have *global* reflectional symmetry, such symmetry occurs on smaller scales, so that stronger equivariance in earlier layers might be helpful.

However, we don't observe clear improvements over pure $C_{16}$-equivariance. A reason could be that even the $C_{16}$-equivariant networks are already very parameter efficient compared to classical CNNs, so that parameter efficiency and equivariance are not a bottleneck anymore. It is also possible that the architecture would need to be adapted slightly to profit from the $D_{16}$-equivariant layers.

---

[11]Note that our non-equivariant performance is significantly worse than the one obtained by Ribeiro et al. (2020)—this is because we randomly rotated the samples, resulting in a more challenging task.

Table 4: Relative equivariance errors for $C_{16}$ on a test image, averaged over 10 random initializations of the layer. As orientation, we also include non-equivariant (vanilla) convolutions.

| Method | Stencil | Error [1e-6] |
|---|---|---|
| Vanilla convolution | $3 \times 3$ | $32716 \pm 5484$ |
| | $5 \times 5$ | $32785 \pm 5620$ |
| Kernels | $3 \times 3$ | $5.0 \pm 1.0$ |
| | $5 \times 5$ | $5.0 \pm 1.3$ |
| FD | $3 \times 3$ | $4.8 \pm 1.1$ |
| | $5 \times 5$ | $6.6 \pm 1.2$ |
| RBF-FD | $3 \times 3$ | $5.9 \pm 1.2$ |
| | $5 \times 5$ | $6.1 \pm 1.1$ |
| Gauss | $3 \times 3$ | $4.9 \pm 1.7$ |
| | $5 \times 5$ | $6.4 \pm 0.8$ |

Table 5: MNIST-rot results with restriction from $D_{16}$ to $C_{16}$ equivariance. Test errors $\pm$ standard deviations are averaged over six runs. See main text for details on the models.

| Method | Stencil | Error [%] | Params |
|---|---|---|---|
| Kernels | $3 \times 3$ | $0.717 \pm 0.022$ | 709K |
| | $5 \times 5$ | $0.710 \pm 0.020$ | 1.1M |
| FD | $3 \times 3$ | $1.248 \pm 0.060$ | 709K |
| | $5 \times 5$ | $1.436 \pm 0.063$ | 947K |
| RBF-FD | $3 \times 3$ | $1.396 \pm 0.059$ | 709K |
| | $5 \times 5$ | $1.565 \pm 0.048$ | 947K |
| Gauss | $3 \times 3$ | $0.806 \pm 0.047$ | 709K |
| | $5 \times 5$ | $0.778 \pm 0.051$ | 947K |

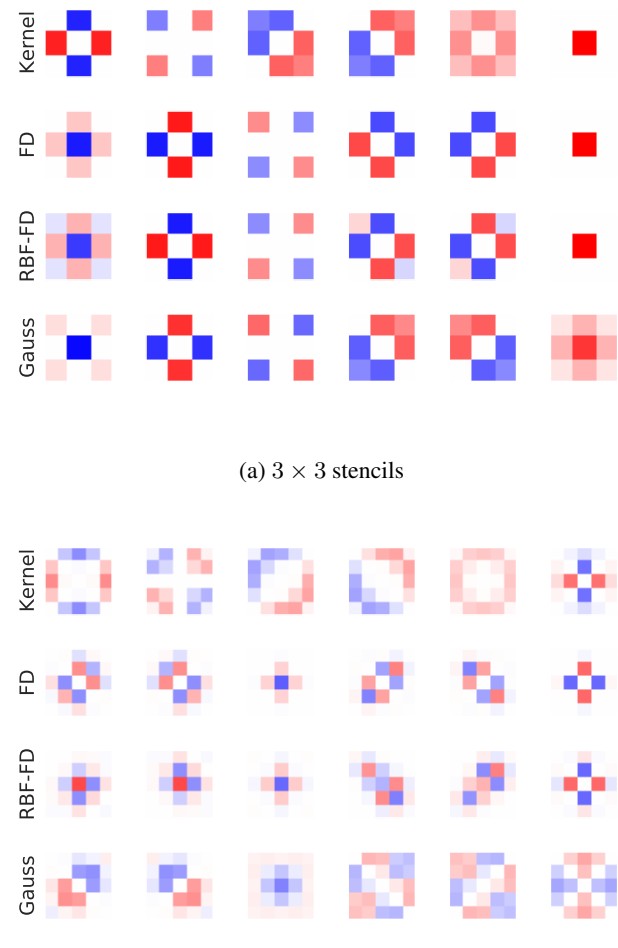

(a) $3 \times 3$ stencils

(b) $5 \times 5$ stencils

Figure 4: Stencils of basis filters for a trivial to regular layer. Each row is a different method and contains six arbitrarily selected filters from the basis (which is not the entire basis). All the settings are those that were actually used for the MNIST-rot experiments.

### N.4 STENCIL IMAGES

Figure 4 contains examples of stencils used during the MNIST-rot experiments. For the $3 \times 3$ stencil, the different methods yield qualitatively similar results (though FD and RBF-FD have fewer stencils that make use of the four corners). But for $5 \times 5$ stencils, kernels and Gauss discretization make significantly more use of the outer stencil points than FD and RBF-FD.

Table 6: Architecture for MNIST-rot experiments

| Layer | Output fields |
|---|---:|
| Conv block | 16 |
| Conv block | 24 |
| Max pooling | |
| Conv block | 32 |
| Conv block | 32 |
| Max pooling | |
| Conv block | 48 |
| Conv block | 64 |
| Group pooling | |
| Global average pooling | |
| Fully connected | 64 |
| Fully connected + Softmax | 10 |

## O    DETAILS ON EXPERIMENTS

### O.1    MNIST-ROT EXPERIMENTS

For the MNIST-rot experiments, we use an architecture similar to one from (Weiler & Cesa, 2019). Table 6 contains a listing of all the layers. Each conv block consists of a $3 \times 3$ or $5 \times 5$ steerable layer, either convolutional or a PDO, followed by batch-normalization and an ELU nonlinearity. The output fields are the number of $C_{16}$-regular feature fields that is used; in the case of the Vanilla CNN and for quotient representations, the number of fields is adjusted so that the parameter count is approximately preserved.

For the quotient experiments, we use $5\rho_{\text{regular}} \oplus 2\rho_{\text{quot}}^{C_{16}/C_2} \oplus 2\rho_{\text{quot}}^{C_{16}/C_4} \oplus 4\rho_{\text{trivial}}$ as the representation, where the numbers are scaled to reach the same parameter count as the model with only regular representations. This combination of representations is the same one used by Weiler & Cesa (2019) and we refer to their appendix for motivation on why we need to combine different representations when using quotients.

We trained all MNIST-rot models for 30 epochs with Adam (Kingma & Ba, 2015) and a batch size of 64. The training data was normalized and augmented using random rotations. For the final training runs, we used the entire set of 12k training plus validation images, as is common practice on MNIST-rot. The initial learning rate was 0.05, which was decayed exponentially after a burn-in of 5 epochs at a rate of 0.7 per epoch. We used a dropout of 0.5 after the fully connected layer, and a weight decay of 1e-7.

These hyperparameters are based on those used in (Weiler & Cesa, 2019); the main difference is that we use another learning rate schedule, which works better than the original one for all models.

For the Gaussian discretization models, we use a standard deviation of $\sigma = 1$ for $3 \times 3$ stencils and $\sigma = 1.3$ for $5 \times 5$ stencils; we chose these values by visual inspection of the stencils, with the aim that full use is made of the stencil (see Appendix N). The RBF-FD discretization uses third-order polyharmonic basis functions, i.e. $\varphi(r) = r^3$.

### O.2    STL-10 EXPERIMENTS

For the STL-10 experiments, we used exactly the same architecture and hyperparameters as Weiler & Cesa (2019), which in turn are essentially those of DeVries & Taylor (2017). This means we train a Wide-ResNet-16-8 (Zagoruyko & Komodakis, 2016) for 1000 epochs, with SGD and Nesterov momentum of 0.9, a batch size of 128 and weight decay of 5e-4. We begin with a learning rate of 0.1 and divide it by 5 after 300, 400, 600 and 800 epochs. For data augmentation, we pad the image by 12 pixels, then randomly crop a $96 \times 96$ pixel patch, randomly flip horizontally, and apply Cutout (DeVries & Taylor, 2017) with a cutout size of $60 \times 60$ pixels.

## O.3 COMPUTATIONAL REQUIREMENTS

We performed our experiments on an internal cluster with a GeForce RTX 2080 Ti and 6 CPU cores. A single run of an MNIST-rot model took about 12 minutes and a run of the STL-10 model about 5.5h. Training the fluid flow prediction model took about 45 minutes. Multiplying this by the number of experiments we did and by six runs with different seeds, the MNIST-rot results took about 34 hours to produce, the STL-10 results about 264 hours, and the fluid flow results about 18 hours. The initial tests we did to debug and find a good learning rate schedule (on MNIST-rot) took much less time than that. So we estimate that producing this paper took around 350 GPU-hours on the GeForce RTX 2080 Ti.

