# OpenReview forum: "Steerable Partial Differential Operators for Equivariant Neural Networks"
_ICLR.cc/2022/Conference — ICLR 2022 Poster_

### Official Review · Reviewer_p5Vs · 2021-11-02

**Correctness:** 4
**Technical Novelty And Significance:** 3
**Empirical Novelty And Significance:** 3
**Recommendation:** 6
**Confidence:** 3

**Main Review:**

Strength:
+ The paper is very rigorously and well-written. The layout of the paper is very clean.
+ The paper tries to bridges the idea between deep learning and physics, drawing attention in particular to convolution between features and PDOs between vector fields.
+ Even though the derivation of the PDO G-steerablility constraint follows similarly from that of kernel steerability constraint, the theory developed is interesting. In particular, it is interesting to see a special example in section 2 that equivariant PDOs are combinations of well-known operators such as Laplacian and divergence.
+ Connections between convolutions and PDOs have also been made through Schwartz distributions in the appendix, which I have to admit that I have not read in full.

Weakness:
- It seems that in Table 1 and 2, equivariant PDOs typically underperform steerable CNNs. The authors did explain the reason being the locality of PDOs, but such underwhelming results do in some sense limit the utility of the developed equivariant PDOs.
- It would be great to see examples where PDOs have a significant advantage over steerable CNNs.

**Summary Of The Paper:**

The authors developed the general theory of equivariant partial differential operators (PDOs) between feature fields on the Euclidean space. Given an arbitrary group G, the G-steerability constraint is derived that fully characterized when a PDO is equivariant between vector fields for given representations. Experiments on the rotated-MNIST and STL-10 datasets have been conducted to compare the performance of equivariant-PDOs and equivariant steerable kernels.

**Summary Of The Review:**

The theoretical contribution of the paper is significant. Even though experimental results are not as satisfactory to demonstrate the utility of equivariant PDOs, I am leaning towards accepting the paper.

---

> ### Author Response · Authors · 2021-11-18
> **Potential future advantages of PDOs over kernels; additional experiments**
>
> Thank you for appreciating our theoretical contributions and the bridge we are helping to build between deep learning and physics! We agree with your points regarding the empirical results: in our experiments, kernels outperform PDOs, and having cases where PDOs really shine would certainly be great. We do believe such cases exist, but unfortunately, they tend to be significant extensions building on top of our work, which we think are out of scope for this particular paper. Perhaps the most promising case are manifolds: kernels on manifolds require a Riemannian structure [1], while PDOs wouldn’t need this and can thus achieve general diffeomorphism equivariance. Their locality could also be an advantage in this setting, since the spatial extent of kernels can be undesirable if the manifold is highly curved in some places. As we briefly mention in our Conclusion, such an extension would still make use of the theory we derive (but would also have to generalize it).
>
> One set of experiments we can add is making use of the capability of steerable PDOs to work with vector fields. Specifically, we can for example predict the flow field of PDE solutions, such as the Navier-Stokes equations for fluid flow. It is not clear whether this will provide any advantage over steerable kernels, but it will distinguish steerable PDOs more clearly from PDO-eConvs and showcase their greater generality. We are currently investigating these experiments in more detail.
>
> [1] Maurice Weiler et al. “Coordinate Independent Convolutional Networks -- Isometry and Gauge Equivariant Convolutions on Riemannian Manifolds”, 2021.

---

> > ### Comment · Reviewer_p5Vs · 2021-11-25
> > **Thank you for the response.**
> >
> > I would like to thank the authors for their response. I believe the theoretical contribution of the paper is significant, and my rating remains.

---

### Official Review · Reviewer_mpGT · 2021-11-02

**Correctness:** 4
**Technical Novelty And Significance:** 4
**Empirical Novelty And Significance:** 3
**Recommendation:** 8
**Confidence:** 3

**Main Review:**

This paper is exceptionally well written and covers an impressive body of work across multiple different domains. Personally, I was extremely impressed with the production quality and thoroughness of the Appendix as well as the ambitious research goal of introducing equivariant PDOs in the main paper. The main technical results are both novel and illuminating and gracefully extends prior work on $E(2)$-CNNs and their theory.

I do not have many strong criticisms regarding the work but rather a few minor questions and comments. The largest of which is that while $G$-steerable equivariant networks guarantee equivariance between layers there exists another body of work that achieves equivariance in a slightly different manner. Specifically, prior works also achieve equivariance in expectation using a Monte-Carlo estimation of the group convolution integral (see [1]). These approaches differ in that they do not treat data as associated vector bundles and also equivariance is achieved over functions on the group $G$ itself necessitating a lifting operation. While this school of thought is different than the $G$-steerable variety as presented in this paper it would be great to harmonize both directions with a discussion on their differences especially when it comes to practical applications. Finally, the paper could be strengthened a bit more if the discussion on the discretization error for the various discretization methods was elevated from the appendix to the main paper. It would be good here to get some numerical quantification of the errors for the datasets and architectures considered as this is an important insight which informs practice.

***Minor Comments***
- I had a bit of a hard time understanding how a group element $g$ acts on a polynomial $x$ in the main paper. Could the authors provide a bit more exposition---if it doesn't already exist---in an appendix?
- The notation in section 2.1 was a bit confusing. Specifically, $c_{\alpha}: \mathbb{R}^d \to \mathbb{R}$ but right after proposition 1. $c_{\alpha} \in \mathbb{R}$. Is this an artifact of picking constants due to enforcing translational equivariance?


[1] Finzi, Marc, et al. "Generalizing convolutional neural networks for equivariance to lie groups on arbitrary continuous data." International Conference on Machine Learning. PMLR, 2020.

**Summary Of The Paper:**

This paper introduces equivariant Partial Differential Operators as a drop in replacement for equivariant kernels in $G$-steerable networks. The paper provides a thorough treatment on the subject matter and gives the necessary equivariant constraints for PDO's in a similar flavor to $G$-steerable kernels. In addition, explicit solutions to these constraints are provided for subgroups of $O(2)$, $O(3)$ and $SO(3)$. Finally, the paper also empirically validates equivariant PDO's on rotation MNIST and STL-10 highlighting important discussion points on the discretization error.

**Summary Of The Review:**

The paper studies equivariant Partial Differential Operators towards building equivariant networks. The work is both novel, interesting, and brings new perspectives to the equivariant networks literature while maintaining an exceptional level of polish in both writing and execution.

---

> ### Author Response · Authors · 2021-11-18
> **We will incorporate your suggestions; small clarifications answering your questions**
>
> Thank you for your very positive comments and for your suggestions! We agree with the points you raise, details below.
>
> > Specifically, prior works also achieve equivariance in expectation using a Monte-Carlo estimation of the group convolution integral (see [1]). These approaches differ in that they do not treat data as associated vector bundles and also equivariance is achieved over functions on the group $G$ itself necessitating a lifting operation. While this school of thought is different than the $G$-steerable variety as presented in this paper it would be great to harmonize both directions with a discussion on their differences especially when it comes to practical applications.
>
> We agree that our discussion of related work on equivariance could stand to be extended. LieConvs [1] are indeed a very interesting alternative to the Steerable CNN viewpoint---not needing to solve the representation theory is certainly appealing. We will make sure to include them in our discussion.
>
> > Finally, the paper could be strengthened a bit more if the discussion on the discretization error for the various discretization methods was elevated from the appendix to the main paper. It would be good here to get some numerical quantification of the errors for the datasets and architectures considered as this is an important insight which informs practice.
>
> Discretization errors are indeed a topic that we should highlight more. We will attempt to make space for this in the main paper (and can certainly mention the key takeaways from appendix I). Regarding numerical experiments, we will add the equivariance error for each method to our results, i.e. how much the discretization errors lead to a violation of the equivariance condition. Roughly, we see that (1) 5x5 stencils tend to have a lower equivariance error than 3x3 stencils (which is in line with the previous steerable CNN literature and with expectations for PDOs), and (2) that the error guarantee of finite difference discretization does not appear to confer an advantage in practice over other discretizations. While the asymptotic nature of the error bound might not make this extremely surprising, it is certainly worth pointing out. Thank you for the suggestion to include these results!
>
> > I had a bit of a hard time understanding how a group element $g$ acts on a polynomial $x$ in the main paper. Could the authors provide a bit more exposition---if it doesn't already exist---in an appendix?
>
> Certainly! To ensure we end up providing the right kind of context, could you please let us know if the following makes things clear? We can think of polynomials in two ways, either as a formal expression, where $x$ is a placeholder for things to be plugged in, or simply as a specific type of functions on $\mathbb{R}^n$---in which case $x \in \mathbb{R}^n$ is simply the argument of that function. When connecting polynomials to PDOs, we take the first perspective and plug in differential operators for $x$. But otherwise, we often take the second perspective. In particular, we consider group elements $g \in GL(n, \mathbb{R})$, so the action of $g$ on $x$ is simply matrix multiplication.
>
> What this means for the expression $p(gx)$ is the following: we can think of the polynomial $p$ as a function $p: \mathbb{R}^n \to \mathbb{R}$, which happens to be a polynomial in the components of $x$. $p(gx)$ can be considered a different function on $\mathbb{R}^n$, still with $x$ as the argument. We could think of it as composing the function $p$ with the group action of $g$ on $\mathbb{R}^n$. Crucially, this combined function is still a polynomial in the components of $x$, just with different coefficients. It is relatively easy to compute the new coefficients based on the coefficients of $p$ and on the matrix entries of $g$, but this isn’t really necessary to derive any of the theory. We’d be happy to describe the calculation of these coefficients in more detail in an appendix if that seems helpful to aid understanding.
>
> > The notation in section 2.1 was a bit confusing. Specifically, $c_\alpha : \mathbb{R}^d \to \mathbb{R}$ but right after proposition 1. $c_\alpha \in \mathbb{R}$. Is this an artifact of picking constants due to enforcing translational equivariance?
>
> Thank you for pointing out this issue in the notation. $c_\alpha \in \mathbb{R}$ is indeed the consequence of translational equivariance---we identify real numbers with constant functions on $\mathbb{R}^n$ here. We think using a different notation for the two might make things even harder to understand, but we will definitely point and explain this clash of notation explicitly.
>
> [1] Finzi, Marc, et al. "Generalizing convolutional neural networks for equivariance to lie groups on arbitrary continuous data." International Conference on Machine Learning. PMLR, 2020.

---

> > ### Comment · Reviewer_mpGT · 2021-11-21
> > **Rebuttal Response.**
> >
> > Thank you for your thoughtful response to my review. I think this covers all my questions and I'm happy to maintain my score on two conditions.
> >
> > - Can you give a taste for the discussion between LieConvs and $G$-Steerable PDOS/Networks here and if time permits update the draft. This I think would very useful to have for the general community.
> > - Regarding the application of $g$ to a polynomial a small illustrative (even contrived) example in the appendix could useful. But this is not mission critical.
> >
> > Overall, unlike the other reviewers I do not mind the small experimental drop in performance as this paper is not intended to get SOTA anyways. It brings other useful tools that I find particularly interesting.

---

> > > ### Author Response · Authors · 2021-11-23
> > > **Revision uploaded**
> > >
> > > Thank you for your response! We have just revised our submission:
> > > - There is now a more detailed related work section at the end of the introduction (page 2), including a discussion of LieConvs. We have decided to present the relation between the group convolutional and steerable frameworks more generally and then discuss how LieConvs fit within this space.
> > > - We have added Appendix C (page 17), which provides some intuition for the group action on polynomials (along the lines of our reply), as well as a worked out example for computing the transformed coefficients.
> > >
> > > For convenience, here is the new discussion group convolutions, steerable CNNs, and LieConvs. Please see the submission for references:
> > >
> > > ---
> > >
> > > Our approach to equivariance follows the one taken by steerable
> > > CNNs (Cohen & Welling, 2017; Weiler et al., 2018a; Weiler & Cesa, 2019; Brandstetter et al., 2021). They represent
> > > each feature as a map from the base space, such as $\mathbb{R}^{d}$, to a fiber
> > > $\mathbb{R}^{c}$ that is equipped with a representation $\rho$ of the point group
> > > $G$. Compared to vanilla CNNs, which have fiber $\mathbb{R}$, steerable CNNs thus
> > > extend the *codomain* of feature maps.
> > >
> > > A different approach is taken by group convolutional
> > > networks (Cohen & Welling, 2016; Hoogeboom et al., 2018; Weiler et al., 2018b). They represent each
> > > feature as a map from a group $H$ acting on the input space to $\mathbb{R}$.
> > > Because the input to the network usually does not lie in $H$, this requires a
> > > lifting map from the input space to $H$. Compared to vanilla CNNs, group
> > > convolutional networks can thus be understood as extending the *domain* of
> > > feature maps.
> > >
> > > When $H = \mathbb{R}^d \rtimes G$ is the semidirect product of the translation group
> > > and a pointwise group $G$, then group convolutions
> > > on $H$ are equivalent to $G$-steerable convolutions with regular representations.
> > > For finite $G$, the group convolution over $G$ simply becomes a finite sum.
> > > LieConvs (Finzi et al., 2020) describe a way of implementing group convolutions
> > > even for infinite groups by using a Monte Carlo approximation for the convolution
> > > integral. Steerable CNNs with regular representations would have to use similar
> > > approximations for infinite groups, but they can instead also use (non-regular)
> > > finite-dimensional representations.
> > > Both the group convolutional and the steerable approach can be applied to
> > > non-Euclidean input spaces---LieConvs define group convolutions on arbitrary Lie groups
> > > and steerable convolutions can be defined on Riemannian manifolds (Cohen et al., 2019b; Weiler et al., 2021)
> > > and homogeneous spaces (Cohen et al., 2019a).
> > >
> > > One practical advantage of the group convolutional approach employed by LieConvs
> > > is that it doesn't require solving any equivariance constraints, which tends to
> > > make implementation of new groups easier. They also require somewhat less heavy
> > > theoretical machinery. On the other hand, steerable CNNs are much more general.
> > > This makes them interesting from a theoretical angle and also has more practical
> > > advantages; for example, they can naturally represent the symmetries of vector field
> > > input or output. Since our focus is developing the theory of equivariant PDOs and
> > > the connection to physics, where vector fields are ubiquitous, we are taking the
> > > steerable perspective in this paper.

---

> > > > ### Comment · Reviewer_mpGT · 2021-11-26
> > > > **Re: Revision uploaded**
> > > >
> > > > Thank you for your detailed response. I am now satisfied with the answers to my questions and I will maintain my rating.

---

### Official Review · Reviewer_BPgr · 2021-11-04

**Correctness:** 4
**Technical Novelty And Significance:** 3
**Empirical Novelty And Significance:** 2
**Recommendation:** 6
**Confidence:** 3

**Main Review:**

The paper provides a very solid theoretical analysis for equivariant PDO. Different from other works on equivariant PDO, this paper enforces more properties of PDO such as divergence and gradient. Also some theoretical findings are useful to me. For example, the locality of PDO is not helpful. The PDO and convolution can be unified into a single framework.

However, I still have several questions and concerns which I will detail below:

- While theoretical findings are interesting, from my point of view, it’s not surprising and novel. For example, PDO is a specific form of convolution. I think this has been revealed by previous works in particular in the context of the diffusion process.

- Regarding the proposed equivariant PDO, I appreciate the deep theoretical analysis. However, I am not very attracted. First, I believe the PDO-eConv should have already revealed these analyses more or less. Second, the equivariant PDO is worse than other works experimentally. In other words, I am not very sure of the significance of the proposed PDO to our community given the limited performance and novelty.

- Experimental analysis is also very limited. What are the advantages of the proposed PDO compared with previous extensive steerable convolution? I recommend having more results.

- Do you have any insight why the performance of the proposed PDO is worse than convolution? Is that potentially because of the isotropic nature of PDO? If so, I recommend the analysis in this direction.


**Summary Of The Paper:**

The paper develops the theory of equivariant partial differential operators -- a steerable PDO which is equivariant under any given symmetry. Interestingly, this work reveals the relation between convolution and PDO by unifying them into a single framework. The work also provides the rigorous theoretical analysis and proof. The paper also provides experimental results to validate its theoretical analysis.

**Summary Of The Review:**

The paper theoretically develops the steerable PDO which enlarges the family of equivariant operators. However, the proposed method is quite limited in terms of performance. Also, the theoretical findings are not very surprising to me given the previous literature. Thus, I doubt the significance of this work to our community. But in case of any misunderstanding, I would like to hear more from the authors during rebuttal.

> Post-discussion: the authors addressed my concerns. So I'm very happy to improve my rating.

---

> ### Author Response · Authors · 2021-11-18
> **Clarifications on the relation to PDO-eConvs and to convolutions**
>
> Thank you for your review and for the questions you raised! We will answer these below and clarify a few points.
>
> > I believe the PDO-eConv should have already revealed these analyses more or less
>
> The core difference between our approach and PDO-eConvs is the following: PDO-eConvs define *one particular* way of using PDOs as neural network layers, and then show that these layers are equivariant for *regular* representations. In contrast, we begin by asking what the space of *all* equivariant PDOs is, for *arbitrary* input and output representations. This is a very different type of problem that requires a different approach, namely deriving and then solving an equivariance constraint, rather than just postulating a certain layer definition.
>
> In practical terms, not restricting ourselves to regular representations means that steerable PDOs are vastly more general than PDO-eConvs. For example, quotient representations (which tend to slightly outperform regular ones, in our experiments as well as in previous ones for kernels), can only be used with steerable PDOs, not with PDO-eConvs. Perhaps more importantly, steerable PDOs can take vector fields as input or output in a natural way, which is not possible with PDO-eConvs (which cannot represent the natural equivariance behavior of vector fields). Related to that, PDO-eConvs do not cover many of the natural equivariant PDOs that occur in physics and other disciplines, such as the gradient, divergence, and curl.
>
> > While theoretical findings are interesting, from my point of view, it’s not surprising and novel. For example, PDO is a specific form of convolution. I think this has been revealed by previous works in particular in the context of the diffusion process.
>
> Let us clarify the relationship between PDOs and convolutions. First and foremost, PDOs can not be written as convolutions with classical functions in a continuous setting. For example, the gradient operator is not represented by convolution with any kernel (and the same holds for all other PDOs higher than zeroth order).
>
> Regarding the diffusion processes you mention, are you referring to the fact that solutions to the diffusion equation can be written as convolutions with an appropriate heat kernel? This is an important point, but different from the way in which we apply PDOs. The solution of a diffusion equation can be written in the form u(x, t) = exp(tΔ) u(x, t=0), where Δ is the Laplacian. And this evolution operator exp(tΔ) can indeed be written either using the exponential of the Laplacian, or as a convolution with a heat kernel. But exp(tΔ) is not itself a PDO (at least not the way we use the term). Rather, it is an *infinite series* of differential operators, which leads to very different properties than those of (finite) PDOs. Studying such infinite series in the context of deep learning could be fruitful (and has already been done for the particularly simple case of the Laplacian, as you mention). In that case, a careful analysis of the relation to convolutions would be necessary. But the PDOs we discuss are separate from convolutions with classical kernels.
>
> Additional complexities are that Schwartz distributions can unify PDOs with classical convolutions, as we show, and that PDOs are often *discretized* as convolutions. We would be happy to go into more detail about these points if anything remains unclear. We are also considering adding a brief section dedicated to discussing the various ways in which PDOs and convolutions are similar and distinct; your feedback about which parts are confusing is very valuable for this!

---

> > ### Author Response · Authors · 2021-11-18
> > **Response Part 2: Performance and locality of PDOs, additional experiments**
> >
> >
> > > The equivariant PDO is worse than other works experimentally [...] Experimental analysis is also very limited. What are the advantages of the proposed PDO compared with previous extensive steerable convolution? I recommend having more results.
> >
> > You are completely right that PDOs underperform steerable kernels in all of our experiments. Our objective with this work is not to develop a method that outperforms steerable kernels on Euclidean data. Our central motivation is instead theoretical: to enable flow of ideas between physics and deep learning, and also to answer the natural question of what the generalization of PDO-eConvs in the Steerable CNN viewpoint is. That being said, we do anticipate practical applications of our work to cases where steerable kernels are not as easily usable. The two most prominent ones are equivariant Probabilistic Numerical CNNs and an extension to manifolds, as we briefly discuss in our conclusion. These applications require the theory we have developed, but they are significant extensions on top of our work, which we believe to be out of scope for this particular paper.
> >
> > One set of experiments we can add is making use of the capability of steerable PDOs to work with vector fields. Specifically, we can for example predict the flow field of PDE solutions, such as the Navier-Stokes equations for fluid flow. It is not clear whether this will provide any advantage over steerable kernels, but it will distinguish steerable PDOs more clearly from PDO-eConvs and showcase their greater generality. We are currently investigating these experiments in more detail.
> >
> > > Do you have any insight why the performance of the proposed PDO is worse than convolution? Is that potentially because of the isotropic nature of PDO?
> >
> > This is an important question, which we asked ourselves while working on this project. We believe the reason is the inherently local nature of PDOs, as we briefly discuss at the end of Section 4. This locality has the effect that PDOs don’t make full use of their stencil. This was our motivation for testing Gaussian discretization, which alleviates the issue significantly. It also underlies our belief that PDOs could be of high practical use in certain scenarios, such as on manifolds, where locality is in fact a boon.
> >
> > We are not sure what you mean by the “isotropic nature” of PDOs. Certain PDOs (such as the Laplacian) are invariant under rotations, but others are not (such as the gradient). In general, we would say that steerable PDOs are no more or less isotropic than steerable kernels.
> >
> > The discussion of locality and its effect on performance is admittedly rather brief at the moment. On reflection, it should be featured more prominently, especially given that it could be an important pointer to direct future work. Rather than folding it into the “Discussion” paragraph, we will give it its own paragraph in the Experiments section.

---

> > > ### Comment · Reviewer_BPgr · 2021-11-27
> > > **Reply to insight of why PDO is worse than convolution**
> > >
> > > By "isotropic nature", I meant -- from viewpoint of the kernel method -- your PDO is an isotropic kernel if I understand correctly. But I agree that "local nature" would be also one of the possible reasons. Thanks for the clarification!

---

> > > > ### Author Response · Authors · 2021-11-29
> > > > **More details on isotropy**
> > > >
> > > > Thank you for engaging with our responses and for adapting your score after our revision!
> > > >
> > > > Just to avoid confusion regarding isotropy: steerable PDOs are generally not isotropic in the sense of being invariant under all rotations. As an example, see the small stencils in the middle of Fig. 2 in the paper (Fig. 4 in the appendix contains a few more). Just like kernels, these stencils follow certain symmetries (enforced by the steerability constraint), but except for a few cases are not simply isotropic.

---

> > ### Comment · Reviewer_BPgr · 2021-11-27
> > **Reply to PDOs and convolution**
> >
> > It would be great if you could add a brief section to the similarity and distinctions from different viewpoints.

---

> ### Author Response · Authors · 2021-11-23
> **Revision uploaded**
>
> We have just revised our submission; in particular, we have added a more detailed section on related work that also discusses DiffusionNet and the way in which this differs from our approach (see our reply). Thank you for bringing up diffusion and that its relation to PDOs should be discussed.

---

> > ### Comment · Reviewer_BPgr · 2021-11-27
> > **Reply to the revision**
> >
> > Thanks for the revision and clarification.  I am happy to improve my rating.

---

### Decision · Program_Chairs · 2022-01-20

**Decision:**

Accept (Poster)

**Comment:**

The paper develops steerable partial differential operator and show how it can be used to build equivariant network. Experimentation on rotated MNIST and STL10 show the merits of the proposed method. Reviewers agreed on the significance of the work and that it brings new perspective on equivariance that would be interesting to the ICLR community. Accept